METHODS AND RESOURCES

# AlphaFold2-guided engineering of split-GFP technology enables labeling of endogenous tubulins across species while preserving function

**Kaiming Xu**[1], **Zhiyuan Li**[2], **Linfan Mao**[3], **Zhengyang Guo**[1], **Zhe Chen**[1], **Yongping Chai**[1], **Chao Xie**[1], **Xuerui Yang**[2], **Jie Na**[3], **Wei Li**[3], **Guangshuo Ou**[1]*

**1** Tsinghua-Peking Center for Life Sciences, Beijing Frontier Research Center for Biological Structure, McGovern Institute for Brain Research, State Key Laboratory of Membrane Biology, School of Life Sciences and MOE Key Laboratory for Protein Science, Tsinghua University, Beijing, China, **2** School of Life Sciences, MOE Key Laboratory of Bioinformatics, Center for Synthetic & Systems Biology, Tsinghua University, Beijing, China, **3** School of Medicine, Tsinghua University, Beijing, China

* guangshuoou@mail.tsinghua.edu.cn

**Data Availability Statement:** All relevant data are within the paper and its Supporting Information files.

## Abstract

Dynamic properties are essential for microtubule (MT) physiology. Current techniques for in vivo imaging of MTs present intrinsic limitations in elucidating the isotype-specific nuances of tubulins, which contribute to their versatile functions. Harnessing the power of the Alpha-Fold2 pipeline, we engineered a strategy for the minimally invasive fluorescence labeling of endogenous tubulin isotypes or those harboring missense mutations. We demonstrated that a specifically designed 16-amino acid linker, coupled with sfGFP11 from the split-sfGFP system and integration into the H1-S2 loop of tubulin, facilitated tubulin labeling without compromising MT dynamics, embryonic development, or ciliogenesis in *Caenorhabditis elegans*. Extending this technique to human cells and murine oocytes, we visualized MTs with the minimal background fluorescence and a pathogenic tubulin isoform with fidelity. The utility of our approach across biological contexts and species set an additional paradigm for studying tubulin dynamics and functional specificity, with implications for understanding tubulin-related diseases known as tubulinopathies.

## Introduction

The microtubule (MT) cytoskeleton serves as a dynamic framework of protein assemblies, endowing cells with both structural robustness and functional plasticity [1,2]. Assembled through the polymerization of tubulin monomers into cylindrical arrays [3], MTs display a behavior known as dynamic instability—a meticulously orchestrated equilibrium between polymerization and spontaneous depolymerization phases [4]. This regulated dynamic is instrumental for fulfilling diverse physiological functions of MTs [5]. Consequently, the high-resolution imaging of MT dynamics is pivotal for dissecting their mechanistic roles in cellular activities.

**Funding:** This work was supported by the National Natural Science Foundation of China (31991190, 31730052, 31525015, 31861143042, 31561130153, 31671444, and 31871352)(https://www.nsfc.gov.cn/) and National Key R&D Program of China (2019YFA0508401, 2017YFA0503501, and 2017YFA0102900)(https://www.most.gov.cn/) to G.O., and the National Natural Science Foundation of China (323B200173)(https://www.nsfc.gov.cn/) to K.X.. The funders had no role in study design, data collection and analysis, decision to publish, or preparation of the manuscript.

**Competing interests:** The authors have declared that no competing interests exist.

**Abbreviations:** aa, amino acid; Dyf, dye-filling defective; EB, end-binding; FRAP, fluorescence recovery after photobleaching; GFP, green fluorescent protein; GSC, germline stem cell; GV, germinal vesicle; HR, homology recombination; IF, immunofluorescence; KI, knock-in; MAP, MT-associated protein; MSA, multiple sequence alignment; MT, microtubule; NGM, nematode growth medium; OE, overexpression; PAE, predicted alignment error; RNAi, RNA interference; StableMARK, Stable Microtubule-Associated Rigor-Kinesin.

Over the past several decades, a myriad of methodologies has been developed to monitor MT dynamics in vivo. Conventional techniques have used microinjection of fluorescein- or rhodamine-labeled tubulins into cells or embryos to measure MT dynamics [5–7]. Furthermore, green fluorescent protein (GFP)-tagged MT-associated proteins (MAPs), such as end-binding (EB) protein or the recent "Stable Microtubule-Associated Rigor-Kinesin" (StableMARK) [8,9], have been rigorously validated for noninvasive and in vivo visualization without disrupting native MT behavior. Complementary to these approaches, computational algorithms for quantification and automated trajectory analysis have been developed, thereby facilitating the acquisition of quantified datasets and imparting real-time measurement of MT dynamics in living systems [8].

Nevertheless, most existing techniques exhibit intrinsic limitations when delving into the functional specificity of tubulin isotypes, an indispensable aspect of MT biology that contributes to their diverse roles in cellular processes [1]. Metazoan genomes encode an array of α- and β-tubulin genes, each exhibiting unique spatial and temporal expression profiles and specialized posttranslational modifications within cellular compartments [1,10]. For instance, the mammalian tubulin isotypes TUBB2 and TUBB3 are predominantly incorporated into neuronal MTs, playing an indispensable role in neurite outgrowth [11]. Conversely, others (for instance, *Caenorhabditis elegans* TBA-5) are more prevalent in cilia and flagella with exclusive localization within specific ciliary segments, contributing to their unique structural and functional roles [12]. This functional specificity of tubulin isotypes is expected to provide important regulatory mechanisms, empowering the MT cytoskeleton to various functionalities [1]. Thus, the real-time interrogation of tubulin isotypes within living cells is foundational for deciphering the functional intricacies of MTs.

Mutations in tubulin-coding genes cause human diseases collectively designated as tubulinopathies such as cortical malformations, manifesting as microcephaly, lissencephaly, or polymicrogyria [13–15]. The symptomatic spectrum of tubulinopathies extends from severe intellectual deficits to nuanced cognitive impairments, accentuating the indispensable role of tubulins in brain development [15]. Also, tubulinopathies encompass ocular and renal anomalies [16]. Notably, many of these mutations are de novo [16–19], underscoring the underestimated impact of tubulinopathies in human diseases. Indeed, AlphaMissense-based predictions revealed that over 80% missense mutations in tubulins (81.9% for α-tubulin TUBA1A and 82.5% for β-tubulin TUBB2B) are likely pathogenic, overwhelming those in KRAS (66.0%) or BRAF (57.7%)—2 hotspot proteins in cancer research (S1A Fig) [20,21]. Comprehensive exploration of the functional specificities among divergent wild-type and mutant tubulin isotypes holds the promise of elucidating the underlying pathophysiological mechanisms [13], thereby facilitating the development of therapeutic strategies. Therefore, there is an imminent requirement to monitor the dynamics of individual tubulin isotypes in both wild-type and pathological forms.

Currently, GFP-tagged tubulins have been widely used to visualize tubulin isotypes across different species [22,23]. However, C-terminal GFP tagging of tubulin impedes its interaction with MAPs or other regulatory elements, whereas N-terminal tagging obstructs the incorporation of tubulin into MT architecture [22,24]. Moreover, endogenous (or knock-in (KI)) GFP labeling of tubulin produces more pronounced disruptions in MT functionality compared to overexpression (OE) [24]. This could be attributed to the functional redundancy of tubulin isotypes and resilience exhibited by endogenous, nontagged tubulins. Nonetheless, ectopic OE of wild-type GFP-tagged tubulin isotypes has been widely adopted across diverse cellular environments, including the nematode [12], mammalian cell lines [25], or mouse embryos [26], with a fraction of GFP-tagged tubulins providing strong fluorescent signals allowing to visualized MTs. Despite those successful applications to mark wild-type MTs, GFP labeling fails to

trace mutated tubulins associated with tubulinopathy with high fidelity. Instead, the visualization of mutated tubulins were accomplished by immunofluorescence (IF) staining in almost all situations [27–31]. Consequently, there is still a lack of unobtrusive, functional imaging of dynamic tubulin isotypes.

Guided by artificial intelligence–driven AlphaFold2 pipeline [32], we developed a novel strategy for the functional labeling of endogenous tubulins while preserving their inherent functionalities. We overcame 3 technical obstacles: the optimal site for tubulin labeling, the selection of fluorescent markers, and the constitution of the linker sequence. Employing the nematode *C. elegans* as a model system, we showed that a 16-amino acid (aa) linker, in conjunction with sfGFP11 from the split-sfGFP system and inserted at the H1-S2 loop of tubulin [33–35], enabled labeling of either α- or β-tubulin without compromising MT assembly, embryogenesis, or ciliogenesis. Extending this technique to tubulins in HeLa cells and murine oocytes, we achieved visualization of MT networks with significantly diminished background fluorescence. This suggests a majority of labeled tubulin is successfully integrated into MT assemblies, thereby implying that our approach is superior to existing technologies. These findings collectively underscore the extensive applicability of our strategy for functional tubulin labeling across diverse cellular milieus and species.

## Results

### AlphaFold2-guided design of functional fluorescence labeling of tubulin

Leveraging the *C. elegans* system—renowned for its suitability for genetic manipulation and live-cell imaging—we carried out initial investigations. The *C. elegans* genome harbors 9 α-tubulin and 6 β-tubulin isotypes [36]. Supposing that regions of low sequence conservation might be more permissive for the addition of an epitope tag, we targeted the H1-S2 loop of the tubulin chain, a notably variable region (S1B Fig). The H1-S2 loop of tubulin protrudes from the internal surface of hollow MTs [37], and previous studies have elucidated that insertion of up to 17 aa into this loop had no apparent functional disruption on tubulins [34]. Therefore, affinity tags have been inserted into this loop for biochemical preparation or IF imaging of individual tubulin isotypes [38–41]. To identify the most appropriate insertion site within this loop, we performed multiple sequence alignment (MSA) for each set of α- and β-tubulin isotypes to acquire the least conserved and most structurally flexible sites amenable for manipulation (S1C and S1D Fig). Through MSA, the gap between the 43rd Gly and 44th Val in α-tubulin TBA-5 and the gap between the 37th Lys and 38th Gly in β-tubulin TBB-2 were the optimal candidates for insertion (S1C and S1D Fig).

Newly synthesized tubulin is captured by the chaperone prefoldin and subsequently delivered to the ring-shaped chaperonin TRiC / CCT complex [42,43], where it undergoes a stringent conformational folding process to form the mature protein. According to recent high-resolution structural analyses of tubulin within the TRiC / CCT complex [44], the incorporation of a full-length fluorescent protein into 55-kDa tubulin may perturb this folding process, yielding improperly folded tubulin. To demonstrate this, we performed Rosetta relax runs for untagged tubulins as well as GFP- or mScarlet-tagged tubulins within TRiC / CCT complex [45,46]. Full-length GFP or mScarlet was inserted directly into H1-S2 loop of α-tubulin TBA-5, which was subsequently deposited into TRiC / CCT complex for a sufficient relaxation of energy. After relaxation following Rosetta relax protocol in designated "rigid" TRiC / CCT complex, untagged tubulins remained relatively intact and structured in the chaperones (S2A Fig). However, either GFP- or mScarlet-tagged tubulins were crushed thoroughly due to steric hindrance created by full-length fluorescent proteins (S2B and S2C Fig), indicating GFP or mScarlet was not eligible for labeling tubulins within the H1-S2 loop. Therefore, we adopted the split-GFP system, inserting a short 16-aa GFP11 fragment directly into the H1-S2 loop,

while coexpressing the complementary GFP1-10 fragment in the sensory neurons of *C. elegans*. Regrettably, this initial attempt yielded no detectable GFP signal at all (S4F Fig).

Next, we harnessed the AlphaFold2 pipeline to optimize the GFP11 insertion tag. According to predictions by AlphaFold2 (Fig 1A), embedding GFP11 within the H1-S2 loop without an adjacent linker precluded its effective interaction with the complementary GFP1-10 fragment, likely due to the rigid topology of GFP11. To address this problem, we extended the GS-linker on both sides of GFP11 to lengths of 6 aa (GS-linker 1), 12 aa (GS-linker 2), or 16 aa (GS-linker 3), in sequential iterations. AlphaFold-based modeling indicated that a linker of at least 12 aa (GS-linker 2) permitted sporadic binding between GFP11 and GFP1-10 (observed in 1 out of 5 independent models), while a 16-aa GS-linker 3 should enable robust binding (observed in 4 out of 5 independent models) (Fig 1A), which was further corroborated by predicted alignment error (PAE) plots (S1E Fig). These predictions were consistent with our live-cell imaging results (S4F Fig), confirming the reliability of our structural predictions. Notably, tubulins tagged by GFP11 with GS-linker 3 possessed intact structures as well as untagged tubulins in the chaperones after sufficient relax runs by Rosetta (S2D Fig), implying the negligible effect of this tag on tubulin folding process. Consequently, a 16-aa GFP11 flanked by 16-aa GS-linker 3 (termed optimized GFP11-i), emerged as the most efficacious insertion tag for the functional labeling of tubulins (Fig 1B). Moreover, AlphaFold2 anticipated that GFP11-i possessed a broad labeling spectrum, effectively marking a broad array of tubulins explored in our studies, including the nematode β-tubulin TBB-2, human α-tubulin TUBA1A, and mouse α-tubulin TUBA4A (Fig 1C). A schematic of AlphaFold2 pipeline-guided design of functional protein labeling was shown in Fig 1D.

## GFP11-i effectively labels endogenous β-tubulins in *C. elegans* embryos

Initially, we evaluated our engineered GFP11-i constructs in labeling endogenous β-tubulin TBB-2 in *C. elegans* embryos. Utilizing CRISPR-Cas9 genome editing technology [47], we integrated either GFP or GFP11-i into the *C. elegans* genome, resulting in all endogenous TBB-2 being tagged. Concurrently, *glh-1::T2A::gfp1-10*, a KI allele of the germline-specific helicase GLH-1, was used to provide endogenous GFP1-10 fragments as the T2A self-split sequence ensured a definitive partition between GLH-1 and GFP1-10 [33]. During early-stage embryonic development, TBB-1 and TBB-2 are the exclusively expressed β-tubulin isotypes [24,36]. Previous studies revealed that RNA interference (RNAi)-mediated depletion of TBB-1 did not induce apparent phenotypic abnormalities including embryonic lethality, suggesting the predominant role of TBB-2 in mitotic MT formation [24].

As the embryos of *C. elegans* contain a mixture of tagged TBB-2 and untagged TBB-1, to gain insight into functional ramifications of different tags on TBB-2, we performed RNAi assay to deplete TBB-1 in *C. elegans* embryos (Fig 2A). Remarkably, both untagged wild-type TBB-2 and GFP11-i-tagged TBB-2 exhibited negligible effects on embryonic lethality [2.7% for untagged wild-type, $N = 293$; 1.6% for TBB-2 (GFP11-i) alone, $N = 253$; 5.3% for TBB-2 (GFP11-i + GFP1-10), $N = 546$]. Conversely, N-terminal GFP::TBB-2 and C-terminal TBB-2::GFP constructs rendered substantial embryonic lethality (76.1% for GFP::TBB-2, $N = 380$; 99.8% for TBB-2::GFP, $N = 549$) (Fig 2B). Moreover, quantification of brood sizes in RNAi-treated wild-type, TBB-2 (GFP11-i), or GFP::TBB-2 animals revealed a significantly reduced brood size for the GFP::TBB-2 variant as compared to its untagged or GFP11-i-tagged counterparts under identical culture conditions [243.4 ± 35.6 for untagged wild-type; 221.0 ± 53.4 for TBB-2 (GFP11-i + GFP1-10); 10.8 ± 8.2 for GFP::TBB-2, $N = 10$ for each group] (Fig 2C and 2D). Collectively, these findings indicated that GFP11-i labeling did not compromise the inherent functionality of β-tubulin TBB-2.

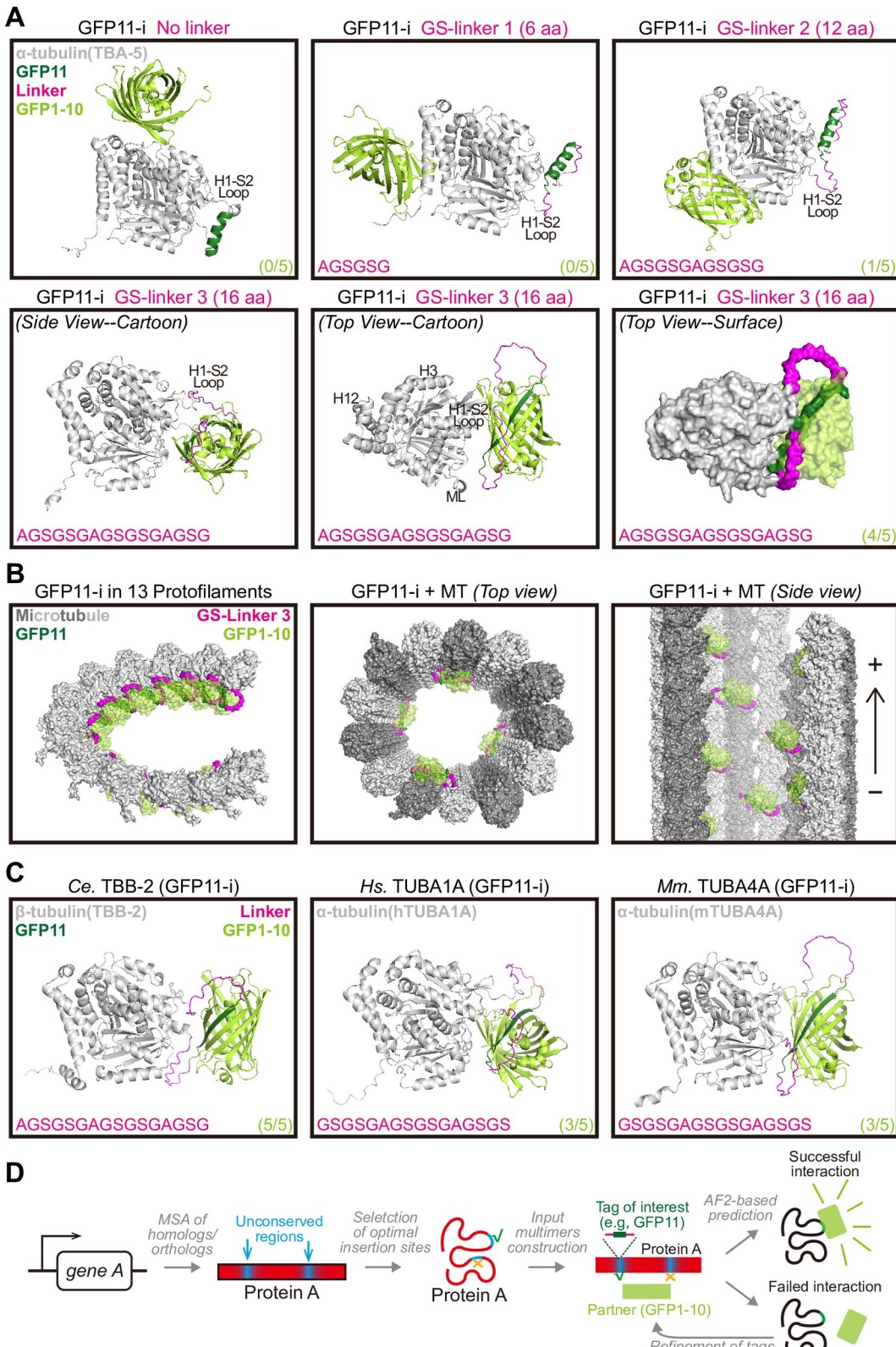

**Fig 1. Models of tubulin (GFP11-i) with GFP1-10 predicted by AlphaFold2.** (**A**) Representative predicted dimer structures of designed tubulin (GFP11-i) flanked by variant linkers and split-sfGFP1-10 (light green). The nematode α-tubulin *Ce*TBA-5 (gray) was employed in these experiments. Split-sfGFP11 (dark green) without linker or flanked by GS-linker 1 (6 aa) / 2 (12

aa) / 3 (16 aa), was inserted into H1-S2 loops of tubulins. Selected insertion sites were indicated in S1B Fig. Sequences of linkers (magenta) on single side were presented on bottom-left of each structure. Ratios of cases with robust interaction between GFP11 and GFP1-10 to all 5 independent cases were presented on bottom-right of each structure. H, helix; S, strand; ML, M loop. (**B**) Modeled tubulin (GFP11-i) positioning in MTs. TBA-5 (GFP11-i) with GS-linker 3 and GFP1-10 dimers in (a) were aligned to tubulins in a MT structural model (PDB: 6U42). (Left) Neighboring 13 tubulins (GFP11-i) in traditional 13-protofilamental MTs. The height difference between the first and last tubulin was 3 units. (Middle) Top view of 4 tubulins (GFP11-i) positioning in MTs. (Right) Side view of 8 tubulins (GFP11-i) positioning in MTs. +, plus end; −, minus end. (**C**) Representative predicted dimer structures of sfGFP1-10 with TBB-2 (GFP11-i) (Left), or hTUBA1A (GFP11-i) (Middle), or mTUBA4A (GFP11-i) (Right). (**D**) Schematic of AF2-guided design of functional tagging of tubulins or other proteins (Protein A). aa, amino acid; AF2, AlphaFold2; MSA, multiple sequence alignment; MT, microtubule.

To further illustrate how endogenous GFP::TBB-2 and TBB-2 (GFP11-i) constructs distinctly affect embryonic development, we used time-lapse microscopy to capture MT dynamics from the first mitotic division (0 min: onset of prometaphase) (Fig 2E and S1 and S2 Videos). In agreement with previous studies [24], GFP::TBB-2 failed to assemble functional spindle MTs, culminating in defective chromosomal segregation into daughter cells, the failure of cytokinesis, and developmental arrest eventually (Fig 2E and S1 Video). In contrast, TBB-2 (GFP11-i) assembled functional spindle MTs, enabling embryos to undergo development that was indistinguishable from wild-type embryos [48] (Fig 2E and S2 Video). These results conclusively demonstrated that GFP11-i enabled endogenous tagging of β-tubulins without engendering discernible functional disruption.

## GFP11-i labels endogenous mutated α-tubulins without affecting phenotypes

Next, we assessed the performance of the GFP11-i labeling approach on structural-sensitive tubulin variants harboring missense mutations, as these variants are analogous to disease-related mutated tubulins [13]. Specifically, we examined a ciliary tubulin isotype, α-tubulin TBA-5, which is predominantly localized in the distal segments of cilia within the sensory neurons of *C. elegans*. Previous studies indicated that several mutations in TBA-5 would disrupt ciliary MTs (Fig 3A) [12]. The A19V missense mutation in TBA-5 was hypothesized to disturb tubulin folding [12,49], as supported by significantly increased van der Waals overlaps (VDWoverlap $\geq$ 0.8 Å) computed by ChimeraX software (S3A Fig) [50]. Remarkably, deletion of TBA-5 (or loss-of-function TBA-5) has negligible effects on cilia due to the functional redundancy of tubulin isotypes [12,36], while A19V mutation transformed the configuration of TBA-5 to "gain-of-function," thereby destructing ciliary MTs upon its assembly into MTs, resulting in the loss of the ciliary distal segments (Fig 3A) [12]. Furthermore, *tba-5 (A19V)* mutants exhibit temperature sensitivity, characterized by the absence of ciliary distal segments at 15˚C, while maintaining relatively intact cilia at 25˚C [12]. This makes the system particularly suitable for investigating whether fluorescent labeling of such tubulin variants has any impact on their functional attributes.

To evaluate functional perturbations arising from fluorescent protein tagging, we engineered multiple *tba-5 (A19V)* constructs with diversified tagging patterns using either full-length GFP or truncated GFP11. These constructs were initially subjected to OE under the control of ciliated neuron-specific P*dyf-1* promoter (S3B Fig) [51]. Consistent with earlier studies [12], wild-type animals possessed intact amphid and phasmid cilia, whereas *tba-5 (A19V)* mutants manifested defective ciliary distal segments when cultivated at 15˚C (S4A and S4B Fig). As expected, OE of untagged TBA-5 (A19V) did not affect defective ciliary architecture observed in *tba-5 (A19V)* mutant (S4C–S4E Fig). Conversely, the C-terminally tagged TBA-5 (A19V)::GFP (GFP-C) with 3xGS linker (S3B Fig) failed to localize within cilia and instead accumulated at the ciliary bases, suggesting C-terminal GFP tagging disrupted the

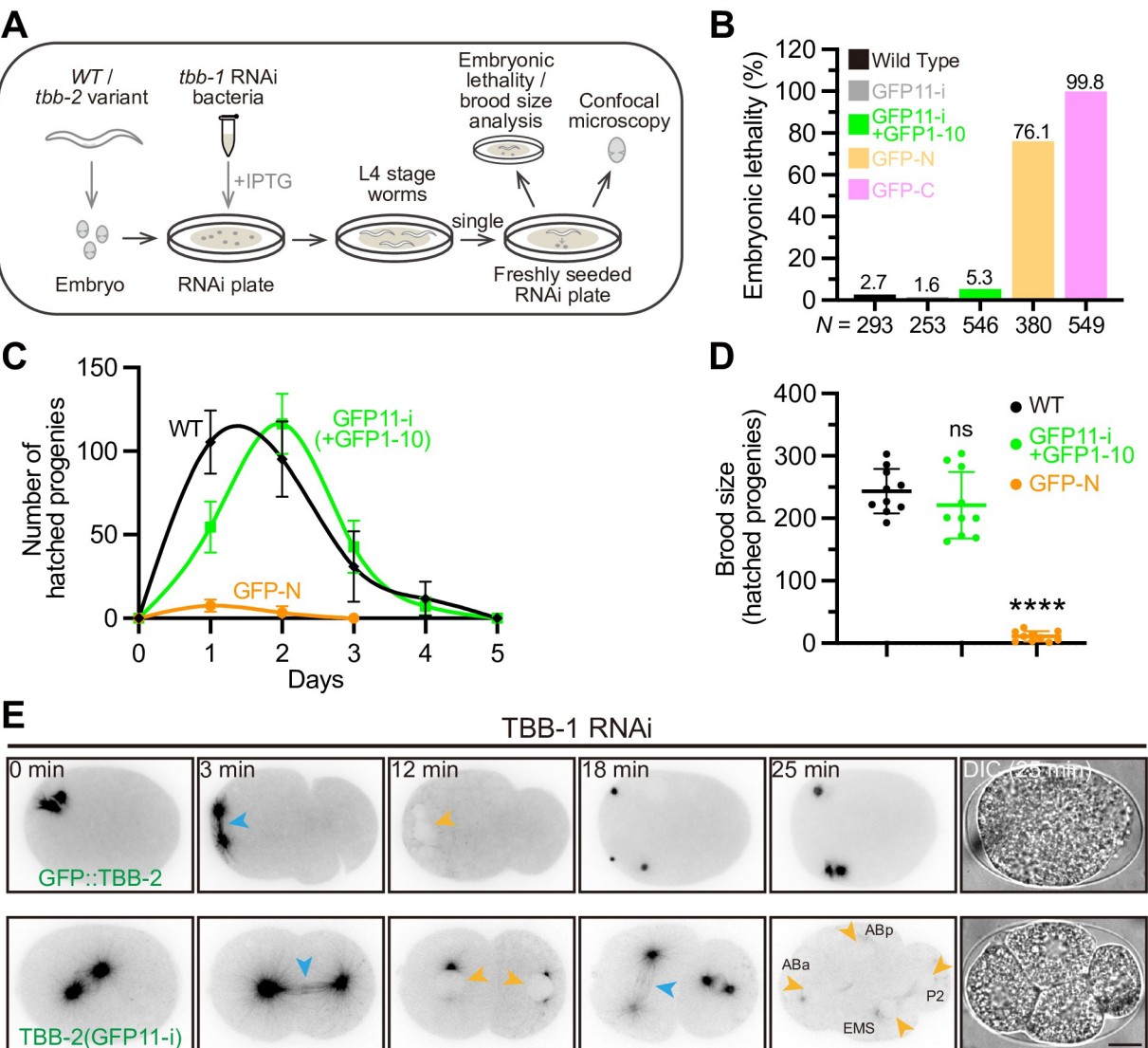

**Fig 2. Functional labeling of endogenous β-tubulin TBB-2.** (**A**) Workflow of *tbb-1* RNAi (knock-down) assay. Target sequence for *tbb-1* could be found in [24]. Fluorescently labeled *tbb-2* variant animals were treated on RNAi plates for one generation before analysis. (**B**) Embryonic lethality in different *tbb-2* variant animals. GFP-N indicated N-terminal GFP::TBB-2, and GFP-C indicated C-terminal TBB-2::GFP. Percentages were shown above the bars of each group. (**C**) Brood sizes of different *tbb-2* variant animals on each day after adulthood (Day 1). The number indicated amounts of hatch worms. Error bars, 95% CI. *N* = 10 for each group. (**D**) Total brood sizes of different *tbb-2* variant animals. Error bars, SD. *N* = 10 for each group. (**E**) Representative images (and stills) for embryonic cleavage in *gfp::tbb-2* or *tbb-2 (gfp11-i)* single-cell zygotes (see also S1 and S2 Videos). "0 min" indicated the onset of prometaphase stage in single-cell zygote. Mitotic spindle MTs were indicated by blue arrowheads. Nuclei were indicated by orange arrowheads. ABa, ABp, EMS, and P2 were daughter cells in 4-cell embryos. Scale bar, 10 μm. Numerical data for panels B-D are available in S1 Data. GFP, green fluorescent protein; MT, microtubule; RNAi, RNA interference.

functionality of tubulin (S4C–S4E Fig, 3 independent OE strains exhibited the same phenotype). Although the N-terminally tagged GFP::TBA-5 (A19V) (GFP-N) localized within cilia, it induced abnormal reconstitution of ciliary distal segments, implicating that N-terminal GFP tagging altered TBA-5 (A19V) function (S4C–S4E Fig). Remarkably, when GFP11-i was inserted into the H1-S2 loop alongside coexpressed GFP1-10 (S3B Fig), the overexpressed *tba-5 (A19V)* variant did not cause discernible morphological alterations in cilia, identical to its untagged counterpart (S4C–S4E Fig). In addition, we examined the fluorescence intensities of

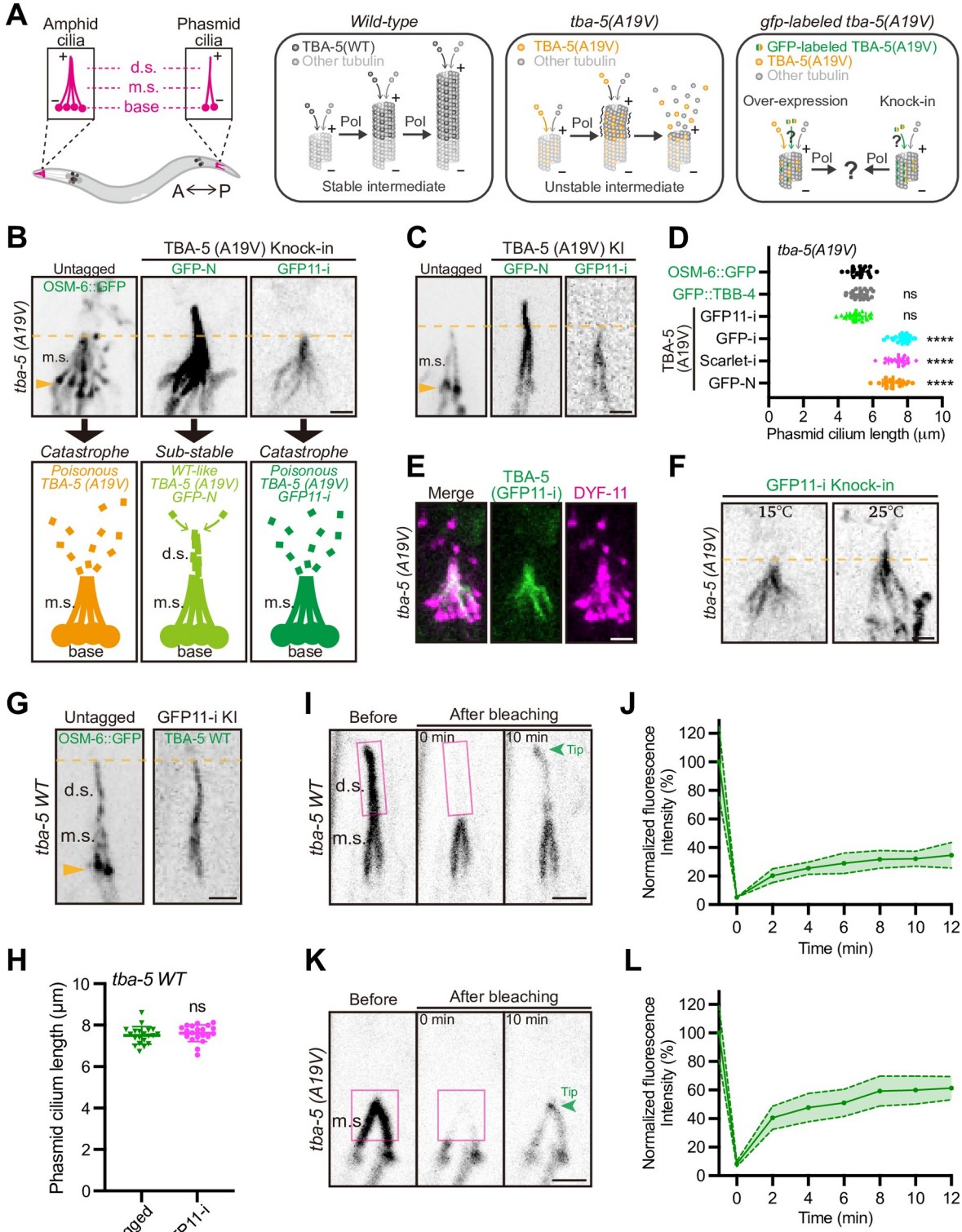

**Fig 3. Functional labeling of endogenous mutated α-tubulin TBA-5 (A19V).** (**A**) Schematic model of ciliary defects caused by mutated TBA-5 (A19V). The amphid and phasmid cilia in *C. elegans* are composed of ciliary bases, m.s., and d.s.. In *wild-type* animals, α-tubulin TBA-5 and other tubulins constitute stable ciliary MTs and generate intact cilia. In *tba-5 (A19V)* mutants, "gain-of-function" TBA-5 (A19V) continuously assembles into plus ends of ciliary MTs, which destabilizes the MT polymer lattice in d.s. and thus destroys the ciliary d.s. [12]. Our study will investigate whether GFP labeling would perturb the structure and functionality of TBA-5 (A19V) and

therefore affect ciliary MTs. A, anterior; P, posterior; Pol, polymerization. (**B** and **C**) Morphologies of amphid cilia (**B**) and phasmid cilia (**C**) when endogenous TBA-5 (A19V) were fluorescently labeled using different strategies. OSM-6::GFP was a component of IFT complex and suitable to mark the defective cilia when TBA-5 (A19V) was untagged. GFP-N indicated N-terminal GFP tag and GFP11-i indicated internal GFP11-i tag. Cartoon models of amphid cilia morphologies were presented underneath each image. Ciliary bases were indicated by orange arrowheads. Scale bar, 2 μm. (**D**) Phasmid cilium length in different *tba-5 (A19V)* variant animals. OSM-6::GFP and GFP::TBB-4 were 2 independent ciliary markers in untagged *tba-5 (A19V)* mutants. $N \geq 18$ for each group. (**E**) Distribution of TBA-5 (A19V) (GFP11-i) in amphid cilia. DYF-11::wrmScarlet (magenta) was a component of IFT complex and marked the cilia morphology. Scale bar, 2 μm. (**F**) Morphologies of amphid cilia in *tba-5 (A19V) (gfp11-i)* animals at 15 or 25°C. Scale bar, 2 μm. (**G**) Morphologies of phasmid cilia when endogenous wild-type TBA-5 was labeled with GFP11-i. OSM-6::GFP marked the wild-type cilia when TBA-5 was untagged. Scale bar, 2 μm. (**H**) Phasmid cilium length in wild-type animals (Untagged) or *tba-5 (gfp11-i)* animals (GFP11-i). (**I**) d.s of phasmid cilia expressing wild-type TBA-5 (GFP11-i) were photobleached (magenta box) at 15°C, and recovery was recorded at 0 min or 10 min. The ciliary tip was indicated by green arrowhead. Scale bar, 2 μm. (**J**) Quantification of FRAP data in *tba-5 (gfp11-i)* animals. The fluorescence intensity was normalized to the prebleach. Data are mean ± 95% CI from 10 independent animals. (**K**) m.s of phasmid cilia expressing TBA-5 (A19V) (GFP11-i) were photobleached (magenta box) at 15°C, and recovery was recorded at 0 min or 10 min. (**L**) Quantification of FRAP data in *tba-5 (A19V) (gfp11-i)* animals. $N = 10$. Numerical data for panels D, J, H, and L are available in S1 Data. d.s., distal segment; FRAP, fluorescence recovery after photobleaching; GFP, green fluorescent protein; IFT, intraflagellar transport; m.s., middle segment; MT, microtubule.

TBA-5 (A19V) tagged with GFP11 flanked by different linkers (S3B Fig). In line with predictions based on Alphafold2 (Fig 1A), only GFP11 flanked by GS-linker 2 (12 aa) or GS-linker 3 (16 aa) enabled stable GFP fluorescence in the presence of GFP1-10, as opposed to GFP11 without linker or with GS-linker 1 (6 aa) (S4F Fig).

To elucidate the functional ramifications of endogenous tubulin labeling, we engineered *tba-5 (A19V)* KI alleles with distinct tagging patterns (S3C Fig). Due to defective ciliary distal segments, *tba-5 (A19V)* animals are unable to uptake the fluorescent dye DiI [52], exhibiting a 100% dye-filling defective (Dyf) phenotype at 15°C (S4G Fig) [12]. The N-terminally tagged GFP KI allele (GFP-N) resulted in abnormal reconstitution of ciliary distal segments and restoration of cilium length (Fig 3B–3D). We varied the linker length between N-terminal GFP and tubulin from 6 aa to 18 aa and observed consistent phenotype. Furthermore, GFP-N KI allele substantially rescued Dyf phenotype, underscoring a significant functional impairment of TBA-5 (A19V) tubulins (S4G Fig).

The *che-3::T2A::gfp1-10*, a KI allele of the ciliated neuron-associated dynein-2 heavy chain CHE-3, served as donor of GFP1-10 fragments in ciliated neurons. Intriguingly, the cilia morphology and length observed in *tba-5 (A19V) (gfp11-i)* KI animals were indistinguishable from those of their untagged counterparts (Fig 3B–3E), albeit with a modest rescue of the Dyf phenotype (S4G Fig). Consistent with *tba-5 (A19V)* mutant phenotype [12], we revealed that *tba-5 (A19V) (gfp11-i)* KI animals also possessed relatively intact cilia at 25°C but were devoid of ciliary distal segments at 15°C (Figs 3F and S5A), preserving the temperature-sensitive nature of TBA-5 (A19V) with respect to ciliary morphology. Additionally, we created wild-type *tba-5 (gfp11-i)* KI animals and did not observe any discernible ciliary defect (Fig 3G and 3H). Using the intraflagellar transport component DYF-11 as markers, we found the velocity of ciliary kinesins in GFP11-i-tagged TBA-5 animals was also indistinguishable from that of wild-type animals (S5B and S5C Fig), implying that GFP11-i labeling may not disturb interaction between MTs and MAPs. Taken together, GFP11-i is an effective tool for the functional labeling of α-tubulin TBA-5, especially applicable to mutated forms.

As GFP11-i is amenable to live-cell imaging, we investigated the dynamics of GFP11-i tagged wild-type TBA-5 or TBA-5 (A19V) through FRAP (fluorescence recovery after photobleaching) experiments. Photobleaching of ciliary distal segments expressing wild-type TBA-5 (GFP11-i) resulted in fluorescence recovery primarily at the ciliary tips, which represented the highly dynamic plus ends of A-tubules of axonemal MTs (Fig 3I and 3J) [12]. The average recovery rate at the ciliary tips was 32.1 ± 7.3% after a 10-min recovery (Fig 3J), comparable to that (approximately 30%) measured in previous studies using overexpressed TBB-4::YFP [12].

Given that TBA-5 (A19V) disrupted ciliary distal segments at 15˚C, we photobleached the remaining ciliary middle segments expressing TBA-5 (A19V) (GFP11-i) and observed fluorescence recovery preferentially at the ciliary tips, which represented the plus ends of both A-tubules and B-tubules of axonemal MTs (Fig 3K and 3L) [12,52]. The average recovery rate at the ciliary tips was 60.0 ± 13.6% after a 10-min recovery in *tba-5 (A19V)* mutant (Fig 3L), faster than that of middle (B-tubule, approximately 25%) or distal segment tips (A-tubule, approximately 30%) measured in wild-type phasmid cilia [12], implying that TBA-5 (A19V) was incorporated into both A- and B-tubules of ciliary MTs. These findings were previously inaccessible using either traditional fluorescent protein tagging or IF imaging, underscoring the exclusive advantage of GFP11-i labeling in exploring the dynamic functionality of tubulin isotypes or variants, a fundamental aspect of MT functionality.

To demonstrate that insertion of full-length fluorescent proteins into H1-S2 loops disrupts the functionality of tubulins as predicted (S2B and S2C Fig), we engineered internally tagged mScarlet or GFP KI allele (Scarlet-i or GFP-i) (S3C Fig). Identical to GFP-N, Scarlet-i resulted in reconstituted ciliary distal segments (S5D and S5E Fig), restored cilium length (Fig 3D), rescued Dyf phenotype (S4G Fig) along with lost temperature-sensitive nature (S5F Fig), indicating a significant structural and functional impairment of TBA-5 (A19V) tubulins. GFP-i exhibited the same phenotype as Scarlet-i, including the reconstituted ciliary distal segments (S5D and S5E Fig) and restored cilium length (Fig 3D). These data suggested the gain-of-function phenotype of TBA-5 (A19V) was suppressed by full-length GFP or mScarlet labeling.

## GFP11-i enables tissue-specific labeling of endogenous tubulins

Despite GFP11-i enables ubiquitous labeling of MTs using ubiquitously expressed GFP1-10, we next investigated if GFP11-i facilitates tissue-specific visualization of endogenous MTs. We genetically crossed the ubiquitously expressed *tbb-2 (gfp11-i)* KI strain with 3 distinct endogenous *gfp1-10* donor elements (Fig 4A). These included ciliated neuron-specific *che-3::T2A::gfp1-10* (*Pcilium*::GFP1-10), hypodermis-specific P*col-19::gfp1-10* single-copy KI (*Phypodermis*::GFP1-10), and germline-specific *glh-1::T2A::gfp1-10* (*Pgermline*::GFP1-10). Consequently, TBB-2 was specifically labeled in the ciliated neurons, the hypodermal cell layer, and the germline along with embryos (Fig 4A).

Using the ciliated neuron-specific labeling, we observed robust distribution of endogenous TBB-2 within amphid and phasmid ciliary MTs (Figs 4B and S6A). Previous attempts using N-terminal GFP::TBB-2 KI allele had masked this phenomenon [22], likely due to background signals emanating from extraneous tissues (S6A Fig). Furthermore, we found that ciliary TBB-2 was predominantly localized within the middle segments of cilia, with diminished signals in the distal segments, thereby revealing a unique and regulated distribution pattern distinct from that of TBB-4 distributing along the full-length cilia [12]. These new findings were previously inaccessible with traditional labeling strategies.

The hypodermis-specific labeling facilitated visualizing dynamic architecture of MT network in the hypodermal cell layers, effectively eliminating signal interference from adjacent tissues. By imaging of hypodermal MTs throughout the nematode, including the head, pharynx, vulva, and tail regions (Figs 4C and S6B), we captured MT dynamics—encompassing polymerization (pol), depolymerization (depol), and translocation (trans) via kymographs and time-lapse videography (Fig 4D and 4E and S3–S5 Videos).

The germline-specific labeling enabled tracking of the temporal behavior of mitotic and meiotic MTs within germline as well as in early-stage embryos, excluding background noise emanating from other unrelated tissues (Figs 4F and S6C). In the mitotic germline, the self-renewing mitosis of individual germline stem cell (GSC) was unambiguously captured, spanning from metaphase to

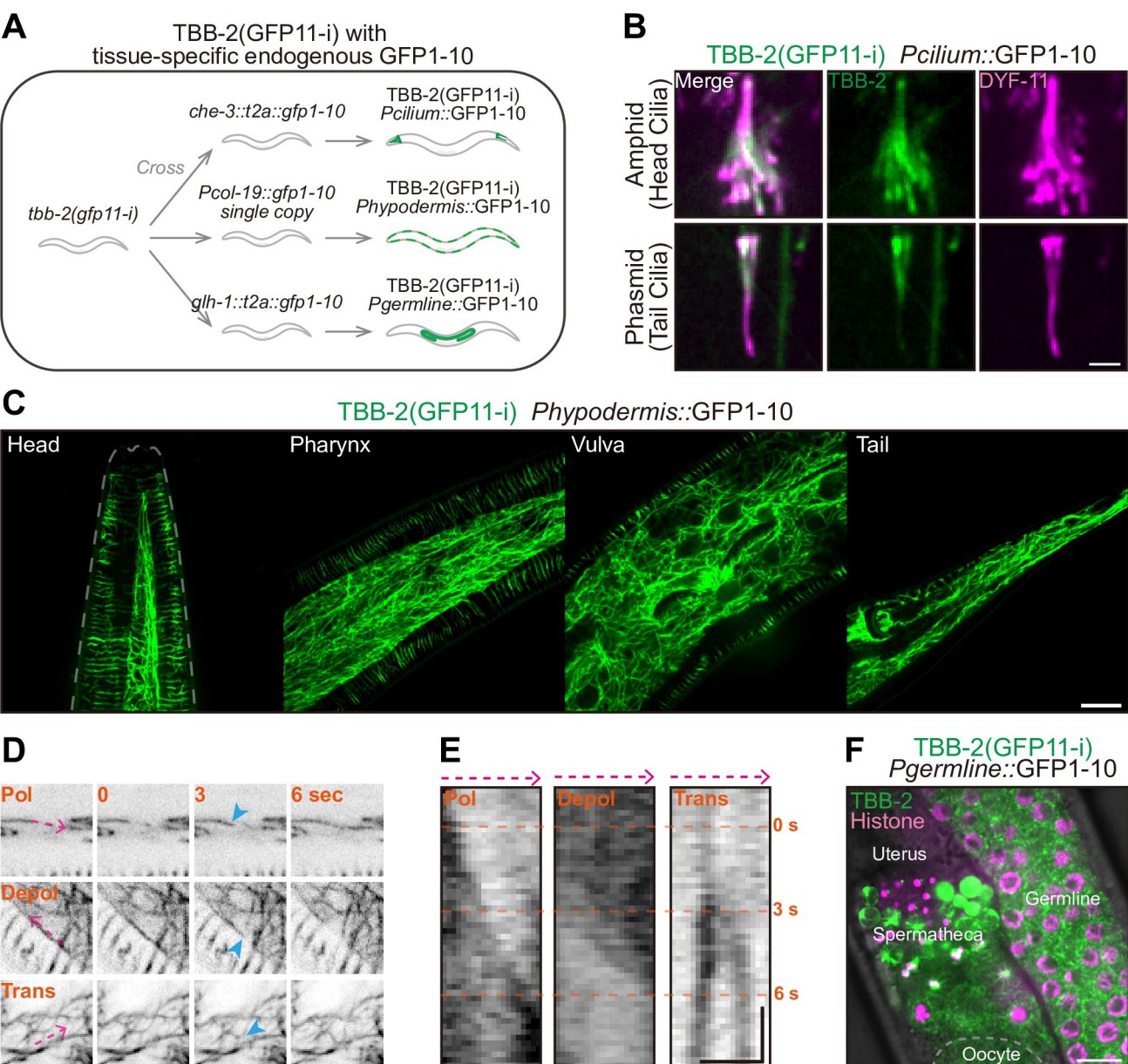

**Fig 4. Tissue-specific labeling of endogenous TBB-2.** (**A**) Flowchart of tissue-specific labeling of endogenous TBB-2 (GFP11-i) using 3 endogenous GFP1-10 donors. *che-3::t2a::gfp1-10* provided GFP1-10 in ciliated neurons, P*col-19::gfp1-10 single copy (KI)* provided GFP1-10 in hypodermis, and *glh-1::t2a::gfp1-10* provided GFP1-10 in germline and early-stage embryos. (**B**) Representative images of endogenous TBB-2 (GFP11-i) with ciliated neuron-specific GFP1-10, in amphid and phasmid cilia. DYF-11::wrmScarlet marked the sensory cilia. Scale bar, 2 μm. (**C**) Representative images of endogenous TBB-2 (GFP11-i) with hypodermis-specific GFP1-10, in head, pharynx, vulva, and tail. Scale bar, 10 μm. (**D**) Stills of MT dynamics in hypodermis (see also S3–S5 Videos). TBB-2 (GFP11-i) with hypodermis-specific GFP1-10 served as MT marker. Pol, polymerization; Depol, depolymerization; Trans, translocation. Dynamic MTs were indicated by blue arrowheads. Scale bar, 5 μm. (**E**) Kymographs of MT dynamics in (D). Kymographs were plotted along dashed lines (magenta) in each image in (D). Scale bar (horizontal), 2 μm; Scale bar (vertical), 2 s. (**F**) Representative images of endogenous TBB-2 (GFP11-i) with germline-specific GFP1-10, in the hermaphrodite germline. Scale bar, 10 μm. GFP, green fluorescent protein; KI, knock-in; MT, microtubule.

telophase (S7A Fig and S6 Video). Likewise, we also monitored the meiosis of primary spermatocytes within hermaphrodite (S7B Fig and S7 Video), the meiosis of fertilized oocytes (S7C Fig and S8 Video), and embryonic cleavage from 1-cell zygote to 4-cell embryo (S7D Fig and S9 Video). All these processes were concordant with previous observations [48,53–55].

To investigate potential effects of GFP11-i on MT dynamics, we quantified astral MT growth rates during 1-cell stage embryonic mitosis. Previous studies employing EB protein EBP-2::GFP as plus-end tip markers revealed astral MT growth rates of 0.72 ± 0.02 μm/s during metaphase and reduced to 0.51 ± 0.02 μm/s in anaphase [56]. Using endogenous EBP-2:: mNeongreen as markers (S8A Fig), we confirmed the rates were 0.68 ± 0.08 μm/s during metaphase and reduced to 0.55 ± 0.06 μm/s ($N = 70$) during anaphase (S8B Fig). By quantifying these profiles using TBB-2 (GFP11-i) in *tbb-2 (gfp11-i)* embryos (S8C Fig), the astral MT growth rates were 0.69 ± 0.09 μm/s during metaphase and reduced to 0.55 ± 0.05 μm/s ($N = 70$) in anaphase (S8D Fig). These rates were commensurate with those measured via EBP-2::mNeongreen, which suggested that GFP11-i labeling may not perturb intrinsic MT dynamics. Collectively, we conclude that GFP11-i offers superior endogenous labeling of both α- and β-tubulin isotypes, outperforming existing fluorescent protein-based labeling methodologies in *C. elegans*.

## GFP11-i enables labeling of mammalian tubulins

To examine the applicability of GFP11-i labeling across multiple species, we tested this technique with the human α-tubulin isotype hTUBA1A, β-tubulin hTUBB8, and mouse α-tubulin TUBA4A (Fig 1D), all of which had been implicated in tubulinopathies [13]. Equivalent quantities of CMV promotor-driven *gfp-htuba1a* or *htuba1a (gfp11-i)* constructs were transiently transfected into HeLa cell lines along with GFP1-10, individually (Fig 5A). Within GFP-positive cells, MTs labeled with GFP11-i exhibited significantly improved signal-to-noise ratio (mean MT / background fluorescence ratio = 3.3, $N = 10$ cells), nearly twice higher than that labeled with GFP (mean MT / background fluorescence ratio = 1.7, $N = 10$ cells) (Fig 5B and 5C). This improvement of the signal-to-noise ratio likely stemmed from the superior functionality of GFP11-i-labeled tubulins, which may facilitate more efficient assembly into MTs. Likewise, we effectively labeled β-tubulin hTUBB8 using GFP11-i tagging strategy (S8E Fig). Importantly, the expression of GFP11-i-labeled hTUBA1A did not change the level of the αK40 acetylation, a posttranslational modification occurred within the lumen of MTs (S8F and S8G Fig) [57], which suggests that the property of the MT lumen may not be strongly affected by GFP11-i labeling.

Recently, several missense mutations in TUBA4A were identified to cause oocyte developmental arrest and human infertility [18,28]. IF staining indicated that mTUBA4A mainly localized within meiotic spindle MTs of mouse oocytes [28], but the behavior of mTUBA4A cannot be visualized in living cells using IF. We thus generated mRNA of GFP-mTUBA4A, mTUBA4A-GFP, and mTUBA4A (GFP11-i) and microinjected these constructs, respectively, with GFP1-10 mRNA into germinal vesicle (GV)-stage mouse oocytes (Fig 5D). We found conventional GFP-tagged mTUBA4A yielded faint fluorescence in meiotic spindle MTs and overshadowed by nonspecific background fluorescence (MT / background fluorescence ratio = 1.1, $N = 6$ cells) (Fig 5E and 5F). This unsuccessful GFP labeling could not be attributed to our artificial or technical limitations, as the GFP-tagged β-tubulin mTUBB5-GFP marked meiotic spindle MTs in mouse oocytes well (S8H and S8I Fig) [26,27]. In striking contrast to GFP-mTUBA4A or mTUBA4A-GFP, GFP11-i permitted improved visualization of spindle MTs, accompanied by significantly attenuated background fluorescence (MT / background fluorescence ratio = 3.0, $N = 6$ cells) (Fig 5E and 5F).

Having demonstrated the effective labeling of wild-type tubulins using GFP11-i, we evaluated whether GFP11-i also allows for labeling of the human pathogenic tubulins with fidelity. The TUBA4A (E284G) de novo mutation was implicated in human infertility, and this pathogenic mutation could led to meiotic arrest in mouse oocytes in a dominant-negative manner

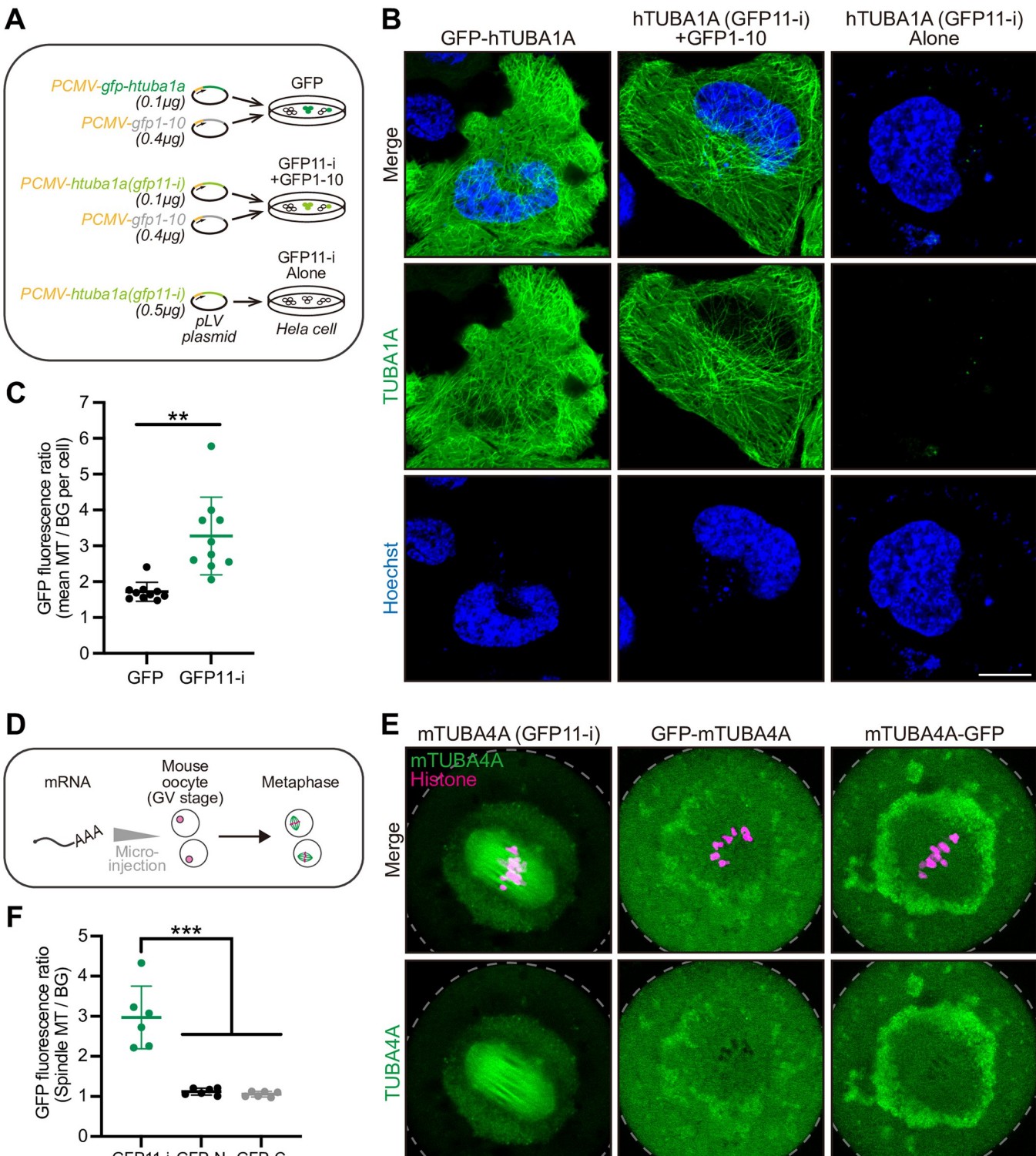

**Fig 5. GFP11-i labeling of mammalian tubulins.** (**A**) Workflow of transgenic expression of GFP or GFP11-i-tagged human α-tubulin TUBA1A in HeLa cells. The total quantity of pLV plasmids for transfection was 0.5 μg in each group. GFP11-i alone served as a negative control. (**B**) Representative images for transfection-positive cells in each group. Hoechst (blue) marked the nuclei of live cells. Images were not z-stacked. Scale bar, 10 μm. (**C**) GFP fluorescence ratio of MTs to their adjacent cytosolic backgrounds. The mean value of 5 independent MTs was calculated in each cell. $N = 10$ independent cells for each group. (**D**) Workflow of mouse α-tubulin TUBA4A mRNA microinjection into GV-stage mouse oocytes. Oocytes at metaphase I were analyzed. (**E**) Representative images for oocytes in metaphase I in each group. Edges of cells were depicted by gray dashed curves. Scale bar, 10 μm. (**F**) GFP fluorescence ratio of spindle MTs to

cytosol backgrounds. The entire meiotic spindle was calculated in each oocyte. *N* = 6 independent oocytes for each group. Numerical data for panels C and F are available in S1 Data. GFP, green fluorescent protein; GV, germinal vesicle; MT, microtubule.

[28]. The residue E284 locates in the M loop of α-tubulin, and, thus, this mutation could potentially attenuate polar lateral interaction with R121 or K124 in the H3 helix of neighboring α-tubulins (Fig 6A). To investigate the impact of E284G on MT stability, equivalent quantities of untagged, or GFP-tagged, or GFP11-i- tagged TUBA4A (E284G) were transfected into HeLa cell lines (Fig 6B). Expression of either untagged TUBA4A (E284G) or GFP11-i-tagged TUBA4A (E284G) disrupted cellular MT arrays (Fig 6B and 6C, GFP as cotransfection marker in untagged group), demonstrating the dominant-negative effect of this mutant. However, the similar expression of GFP-TUBA4A (E284G) failed to induce an apparent defect, likely due to its ineffective incorporation into MT network (Fig 6B and 6C). These findings underscored the fidelity of GFP11-i labeling for live-cell imaging of tubulin isotypes in pathological contexts.

## Discussion

Our study presents advances in the functional fluorescence labeling of tubulins, addressing the long-standing challenge of preserving functionality while achieving robust labeling and real-time imaging. Utilizing the *C. elegans* model and AlphaFold2 pipeline, we successfully engineered an optimized GFP11-i construct that enabled both functional and fluorescent

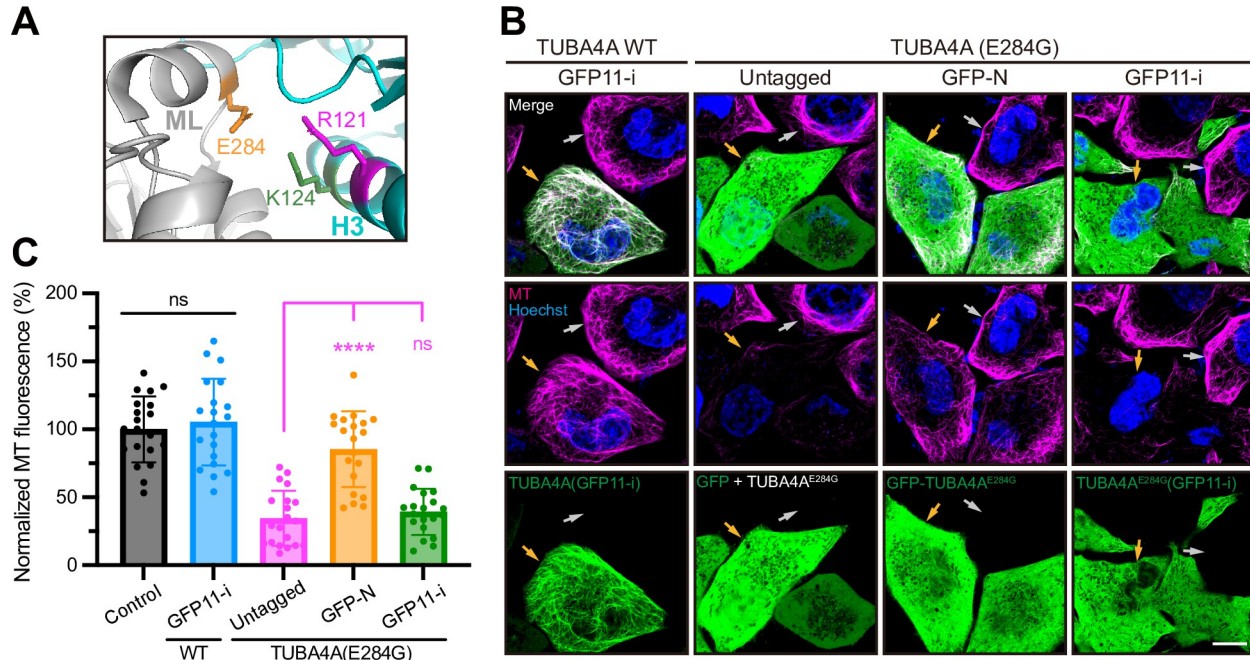

**Fig 6. GFP11-i labeling of disease-related human tubulins.** (**A**) Structural illumination of E284 (orange) in the lateral interaction between 2 neighboring α-tubulins (gray and cyan). ML, M loop; H3, helix 3. (**B**) Representative images for transfection-positive cells (indicated by orange arrowheads) and transfection-negative cells (gray arrowheads) in each group. For untagged group, GFP served as cotransfection marker and untagged TUBA4A (E284G) was expressed in most GFP-positive cells. MTs (magenta) were stained by live-cell tubulin tracker (see also Materials and methods). Hoechst (blue) marked the nuclei of live cells. Scale bar, 10 μm. (**C**) Whole-cell average MT fluorescence in each group of cells. Control represented transfection-negative cells and the average value of this group was normalized to 100%. For other transfection-positive groups, only cells with high expression [as indicated by orange arrowheads in (A)] were measured. *N* = 20. Numerical data for panel C are available in S1 Data. GFP, green fluorescent protein; MT, microtubule.

characterization of endogenous tubulins. By expanding the utility of GFP11-i to different mammalian tubulin isotypes, we validated its applicability in both human cell models and murine embryonic systems. This underscores its considerable promise for understanding MT dynamics across various biological contexts, encompassing both normal physiology and tubulin mutation-associated pathologies.

Previous studies indicated that insertion of up to 17 aa (for instance, 6xHis tag) into the H1-S2 loop of tubulins did not disrupt the functionality of tubulins [34]. Therefore, inserting other small tags into the H1-S2 loop would be also feasible for tubulin purification. However, it remained mysterious how a longer insertion tag would affect the functionality of tubulins. Our initial attempts employing 16-aa GFP11 tags yielded no fluorescence (S4F Fig). It was only after harnessing the predictive capacity of AlphaFold2 that we were able to navigate these pitfalls by optimizing linker length for GFP11 incorporation (Fig 1A). This computational approach, previously untapped for tubulin engineering, provides additional insights into recombinant protein design and paves the way for similar strategies in labeling other structurally complex proteins.

In *C. elegans*, our RNAi assays and quantification of brood sizes substantiated the functionality of GFP11-i labeled TBB-2 (Fig 2A–2E). The data from GFP11-i starkly contrasted with embryonic lethality induced by conventional GFP tagging at either terminus, thereby affirming the pivotal advantage of GFP11-i in preserving protein functionality. The central thrust of our study established GFP11-i as a potent tool for the functional labeling of endogenous mutated α-tubulins without perturbing their molecular activities. In particular, we have investigated the structural and functional impact of GFP11-i labeling on the A19V mutant of the ciliary tubulin isotype TBA-5 in *C. elegans*. Despite structural sensitivity of TBA-5 (A19V) and its impairment of ciliary distal segments, GFP11-i labeling did not induce or exacerbate discernible morphological alterations, including the temperature-sensitive nature, highlighting its promise as an ideal labeling strategy for high-fidelity biological studies. Considering that pathogenic tubulins tend to lose their toxic effects on MTs when labeled with GFP (Fig 6), our developed method enables following the dynamic behavior of pathogenic tubulins in living organisms or cells while preserving their "toxicity" on MTs, thereby holding the promise to advance mechanistic understanding of ciliopathies and other tubulin-related diseases involving tubulin isotypes [13,15].

Our work has broader implications for the fields of cell biology and bioengineering. Beyond tubulins, our method can potentially be adapted to other proteins that are difficult to label without compromising their native functions. The success of our AlphaFold2-guided design provides a framework for leveraging computational tools in the de novo design of functional protein tags. It is tempting to speculate that similar pipelines could be employed to design novel probes for a multitude of applications, ranging from live-cell imaging to targeted therapeutics.

While our study offers promising insights, it also has some potential limitations. Whether GFP11-i in the MT lumen affects recruitment of intraluminal MAPs remains to be demonstrated [1,57]. Although GFP11-i labeling did not compromise MT growth rates during embryonic mitosis in *C. elegans* (S8D Fig), additional parameters such as the catastrophe rates of MTs needs to be identified to further confirm the functionality of GFP11-i labeling. Moreover, a comparative study involving additional labeling techniques, isotypes, and species could further substantiate our conclusions. Future studies are required to explore generating cell lines or mouse models expressing GFP11-i labeled tubulins to investigate MT dynamics in various MT-based processes, such as mitosis or ciliogenesis.

In summary, our GFP11-i labeling technology represents an important advance in the pursuit of efficacious yet minimally perturbative labeling of endogenous proteins. It exhibits a

higher signal-to-noise ratio and promotes MT assembly with remarkable efficiency. Importantly, the GFP11-i strategy enables the visualization of pathogenic tubulin dynamics with fidelity in living cells, thereby providing a substantial advantage over traditional GFP labeling strategies. The broad applicability, corroborated by its successful deployment in *C. elegans* as well as mammalian cells involving both wild-type and structural sensitive mutated tubulin isotypes, renders it suitable for investigations focused on elucidating the intricate dynamics of tubulin (and, potentially, other proteins) in both physiological and pathological contexts.

# Materials and methods

## Worm strains, culture, and genetic cross

*C. elegans* were cultured at 20˚C on standard nematode growth medium (NGM) plates and seeded with the *Escherichia coli* OP50 unless described otherwise. The wild-type strain was Bristol N2. Some strains were provided by the Caenorhabditis Genetics Center (CGC), and other strains were created using the CRISPR-Cas9-mediated genome editing technology. Strain SHX2025 was a gift from Dr. Suhong Xu (Zhejiang University, P.R. China). Strain DUP228 was a gift from Dr. Wei Zou (Zhejiang University, P.R. China). S1 Table summarizes the strains used in this study. For genetic cross, 20 μL suspended OP50 was dropped at the center of NGM plates to make a cross plate. Around 10 to 15 males carrying *him-5 (e1490)* allele and 5 hermaphrodites were transferred to this plate for at least 24 h. F1 and F2 hybrid progenies were singled and screened by PCR using EasyTaq 2× Super Mix (TransGen Biotech, #AS111-14).

## Molecular biology

For KI plasmids, 20 bp CRISPR-Cas9 targets (S2 Table) were inserted into the pDD162 vector (Addgene #47549) by linearizing this vector with 15 bp overlapped primers. The resulting PCR products were treated with *DpnI* digestion overnight and then transformed into *E. coli Trans5α*. The linearized PCR products were cyclized by spontaneous recombination in bacteria. The homology recombination (HR) templates were constructed by cloning the 0.9 to 1.2 kb upstream and downstream homology arms into pPD95.77 vector (Addgene #37465) using In-Fusion HD Cloning Kit (Takara Bio, #639650). Subsequently, fluorescence tag coding sequences were directly inserted into their respective positions in the plasmids. Ultimately, CRISPR-Cas9-targetted sites in the templates were modified with synonymous mutations. For OE plasmids, cDNA construct of *tba-5*, *tba-5 (qj14)*, or *gfp1-10* was cloned into pDONR vector containing 445 bp *dyf-1* promoter and *unc-54* 3′ UTR. cDNA construct of human *tuba1a* or *gfp1-10* was cloned into pLV vector containing CMV enhancer and CMV promotor. cDNA construct of mouse *tuba4a* or *gfp1-10* was cloned into pET27b vector downstream of T7 promotor. Fluorescence tag coding sequences were inserted into their respective positions in those plasmids above. All plasmids were purified with AxyPrep Plasmid Purification Miniprep Kit (Axygen, #AP-MN-P-250) and PureLink Quick PCR purification Kit (Invitrogen, #K310001). mMESSAGE mMACHINE T7 Ultra Kit (Life Tech, #AM1345) was used for in vitro transcription of mouse *tuba4a* or *gfp1-10* mRNAs, and MEGA-clear kit (Life Tech, #AM1908) was used for mRNA purification before microinjection. S3 Table summarizes the plasmids and primers used in this study.

## Genome editing

CRISPR-Cas9-mediated genome editing was used to create KI strains [47]. Target sequences were selected by the CRISPR design tool (https://zlab.bio/guide-design-resources). The CRISPR-Cas9 constructs with targets and HR templates were coinjected into the gonads of

young adult worms at 50 ng/μL with the *rol-6 (su1006)* selection marker (50 ng/μL). F1 transgenic progenies (roller) were singled and screened by PCR. All KI alleles were verified by the Sanger sequencing of the entire genes to guarantee that no other mutations were introduced. SunyBiotech generated some KI strains and alleles in this study, the name of these strains started with "PHX." The name of these alleles started with "syb."

## Extrachromosomal transgenesis (overexpression)

Extrachromosomal transgenic lines of *C. elegans* were obtained by coinjecting the OE plasmids at 10 ng/μL with *rol-6 (su1006)* marker (20 ng/μL) into the gonads of young adult worms. *tba-5 (A19V) (gfp11-i)* (10 ng/μL) and *gfp1-10* (10 ng/μL) constructs were coinjected with *rol-6 (su1006)* into the gonads of worms. At least 2 independent transgenic lines with a constant transmission rate (>50%) were examined and analyzed.

## Transfection and immunofluorescence

HeLa cells were grown in DMEM (Gibco) containing 10% fetal bovine serum (ExCell Bio) and 1% penicillin and streptomycin (Yeasen) at 37˚C with 5% $CO_2$. HeLa cells were seeded on 35 mm glass-bottom confocal dishes the day before transfection. In Fig 5B, cells were transfected with *gfp-htuba1a* (100 ng) and *gfp1-10* (400 ng), or *htuba1a (gfp11-i)* (100 ng) and *gfp1-10* (400 ng) (also in S8F Fig), or *htuba1a (gfp11-i)* (500 ng), respectively, using Lipofectamine 2000 Transfection Reagent (Invitrogen) following manufacturer's instruction. In Fig 6A, cells were transfected with wild-type *htuba4a (gfp11-i)* or differentially tagged *htuba4a (E284G)* (300 ng), with *gfp1-10* (450ng). For untagged group, *htuba4a (E284G)* (300 ng) and *gfp* (100 ng) were cotransfected. In S8E Fig, cells were transfected with *htubb8 (gfp11-i)* (100 ng) and *gfp1-10* (200 ng). Cell culturing medium were renewed 6 h after transfection. The cells were incubated with DMEM medium containing 2 μg/mL Hoechst 33342 and 1× Tubulin Tracker Deep Red (Invitrogen, #T34076) at 37˚C for 30 min before imaging. For IF imaging (S8F Fig), anti-alpha tubulin (acetyl K40) firstantibody (Abcam, #ab179484) (1:300 diluted) and Dylight 549 second antibody (Abbkine, #A23320) were used to stain acetylated αK40, and whole-cell average fluorescence was measured for each cell.

## Mouse oocyte collection and mRNA microinjection

The protocol for mouse oocyte collection was approved by the Institutional Animal Care and Use Committee and Internal Review Board of Tsinghua University. GV oocytes were collected from the ovary of 3 to 4 wk old C57BL/6 females 48 h after PMSG injection and then were incubated in M2 medium (Sigma, #M7167) supplemented with 10 μM Cilostamide (Target-Mol, #68550) at 37˚C with 5% $CO_2$. *tuba4a-gfp* (or *gfp-tuba4a*, or *tubb5-gfp*) and *h2b-mcherry* mRNAs were coinjected at concentrations of 100 ng/μL and 50 ng/μL into the mouse GV oocytes using Eppendorf FemtoJet and a Leica microscope micromanipulator. For GFP11-i, *tuba4a (gfp11-i)*, *gfp1-10* and *h2b-mcherry* mRNAs were coinjected at concentrations of 100 ng/μL, 200 ng/μL, and 50 ng/μL.

## Sequence alignment

Sequence alignment was performed using Clustal X2.1 (http://www.clustal.org/) and visualized through Jalview 2.11.2.1. Protein sequences (TBA-1 to TBA-9; TBB-1 to TBB-6) were obtained from Wormbase (http://www.wormbase.org/). The full-length sequences were used to perform alignment, and conservatism was evaluated by Jalview 2.11.2.1 with default settings.

## Dye-filling assay

The fluorescence dye DiI (1,1′-dioctadecyl-3,3,3′,3′-tetramethylindocarbocyanine perchlorate, Sigma) filling assay was broadly used to assess the ciliary function and integrity [52,58]. Dye-filling positive animals possess relatively intact ciliary structures, while dye-filling defective animals develop abnormal ciliary structures. Worms with *tba-5 (qj14)* background were cultured at 15˚C for at least 24 h before this assay [12]. Young adult worms were harvested into 500 μL M9 buffer with DiI (4 μg/ml), followed by incubation at 15˚C in the dark for 1 h. Worms were then transferred to seeded NGM plates and examined for dye uptake 2 h later using a fluorescence compound microscope. We observed at least 100 animals of each strain from 3 independent assays.

## AlphaFold2-based structural modeling

Tubulin sequences were obtained from National Center for Biotechnology Information (www.ncbi.nlm.nih.gov/). Sequence IDs used include NP_492668.1 (*Ce*TBA-5), NP_497806.1 (*Ce*TBB-2), NP_006000.2 (*Hs*TUBA1A), and NP_033473.1 (*Mm*TUBA4A). Subsequently, sequences were modified through insertion of split-sfGFP11 with variant GS-linkers into H1-S2 loops of tubulin chains [33]. Gaps between G43 and V44 in TBA-5, or between K37 and G38 in TBB-2, or between G43 and G44 in hTUBA1A, or between G43 and G44 in mTUBA4A were candidate insertion sites. Tubulin monomer, or tubulin (GFP11-i): sfGFP1-10 dimer structures were predicted by Alphafold v2.2.0 [32]. The representative model with PAE plot was selected only when it represented the majority (≥3) of the 5 models. To fit GFP1-10-bound tubulin (GFP11-i) into 13 protofilaments or MTs (Fig 1B), MT structural model (6U42 in PDB database) was downloaded as template [59]. Tubulin (GFP11-i) was aligned to single template tubulin in 6U42 using the "Alignment" function of Pymol 2.5.5 (https://pymol.org/2/). This alignment of tubulins allows for correct positioning of GFP11 and GFP1-10 in MTs. TBA-5 (A19V) molecular graphics were constructed using UCSF ChimeraX (https://www.rbvi.ucsf.edu/chimerax/) [60]. The effect of amino acid substitution was modelled using the "Structure Editing/Rotamer" function of ChimeraX and the highest probability of conformation was shown. To find clashes, pairs of atoms with VDW overlap ≥ 0.8 Å were identified using the "Find Clashes/Contacts" function, and 0.4 Å were subtracted from the overlap for potentially H-bonding pairs.

## Rosetta relax runs

Structural model of tubulin within TRiC / CCT complex (7TUB) was downloaded from RCSB PDB database [44]. *Ce*TBA-5 (NP_492668.1), TBA-5 (GFP11-i), TBA-5 (GFP-i), and TBA-5 (Scarlet-i) were predicted by AlphaFold v2.2.0 to acquire their structures. These structures were aligned to template tubulins in 7TUB using the "Alignment" function of Pymol 2.5.5 and then substituted template tubulins. For each model generated after substitution, we run the Rosetta relax protocol multiple times to relieve the clashes between substituted tubulins and neighboring CCT subunits [46]. CCT subunits were set to be rigid in each relax script. Relaxed structures with the highest scores assessed by Rosetta Energy Function 2015 (REF2015.wts) were analyzed and presented [61].

## Live cell imaging

*C. elegans* hermaphrodites were anesthetized with 0.1 mmol/L levamisole in M9 buffer, mounted on 3% agarose pads, and maintained at 20˚C unless described otherwise. Animals with *tba-5 (qj14)* background were maintained at 15˚C during imaging unless described otherwise. *C. elegans* embryos were collected from dissected hermaphrodite adults in M9 buffer and

mounted on 3% agarose pads. For the nematode *C. elegans*, imaging was performed using an Axio Observer Z1 microscope (Carl Zeiss) equipped with 488 and 561 laser lines, a Yokogawa spinning disk head, an Andor iXon + EM-CCD camera, and a Zeiss 10×/0.25 or 100×/1.46 objective. Images were acquired at 0.8-μm z-spacing or 200 ms intervals for 100× objective, or 5-μm z-spacing for 10× objective by μManager (https://www.micro-manager.org). Images were taken using identical settings (EM-gain: 250, exposure time: 200 ms, 30% of max laser). To monitor dynamics of astral MTs in *C. elegans* embryos, the exposure time was 400 ms for each frame. Image stacks in Figs 2E, 3I, 3K, 4B, 4C, 4D, 4F, S4F, S7A, S7B, S7C, S7D S8A, and S8C were z-projected using "average" projection. Image stacks in Figs 3B, 3C, 3E, 3F, 3G, S4A, S4B, S4C, S4D, S5D, S5E, S6A, S6C, S8H, and S8I were z-projected using "max" projection. Stacked images covering the whole cilia structures were used to measure the cilia length. For HeLa cells, imaging was performed using Zeiss LSM900 with Airyscan2 confocal microscopy (Carl Zeiss) equipped with a 63×/1.4 objective. For mouse oocytes, imaging was performed using Zeiss LSM980 with Airyscan2 microscopy (Carl Zeiss) equipped with a cell culture chamber (37°C, 5% $CO_2$) and a 63×/1.4 objective. Images were acquired at laser wavelength of 488 nm for GFP, 594 nm for mCherry, and 405 nm for Hoechst channel. Images were taken using identical settings (Airyscan mode: Airyscan SR; Scan direction: bidirectional; Scan mode: frame; Pixel time: 14.70 μs for HeLa cells or 21.19 μs for mouse oocytes). All images above were not deconvolved. Image analysis and measurement were performed with ImageJ software (http://rsbweb.nih.gov/ij/). The visualized color of the L-561 / 594 channel was changed from red to colorblind-safe magenta in ImageJ. L-488 channels in Figs 2E, 3B, 3C, 3F, 3G, 3I, 3K, 4D, 4E, S4A, S4B, S5D, S5E, S8A, and S8C were changed from green to gray and then inverted fluorescence values in ImageJ. All images shown were adjusted linearly. Kymographs were generated with the KymographClear toolset plugin in ImageJ by manually drawing lines along MTs. The growth rates of astral MTs were calculated using the following function: *Growth Rate = COT(RADIANS(A)) \* $L_{pixel}$ / T*, where *A* is the slant angle in kymograph, $L_{pixel}$ (μm) is the actual width of one pixel, *T* (s) is the interval of frames.

## FRAP

FRAP experiments were carried out at 15°C using Zeiss LSM900 with Airyscan2 confocal microscopy (Carl Zeiss). A 488-nm laser at 100% power was used for photobleaching, and images were acquired every 2 min. We measured fluorescence intensities at the ciliary tips before or after the bleach. The data were normalized to the fluorescence before the bleach.

## RNAi

RNAi was carried out as in [62]. Designed target sequence for *tbb-1* could be found in [24]; this target was proved to inhibit expression of *tbb-1* efficiently. Worms were fed with RNAi bacteria for at least one generation before analysis. To calculate embryonic lethality, 15 day 1 adult worms were transferred to freshly seeded RNAi plates to lay eggs for 1 to 2 h. Then, all adult worms were removed and the number of progenies was counted. Embryonic lethality was calculated by the ratio of unhatched progenies after 3 d to all progenies at the beginning. To calculate brood size, L4 worms were transferred singly to freshly seeded RNAi plates. Worms were repeatedly transferred to freshly seeded RNAi plates every 24 h until reproduction ceased. We counted the number of hatched progenies at their L4 stages.

## Quantification and statistical analysis

The sample sizes in our experiments were determined from the related published analyses. All experiments were repeated at least 2 times from independent samples with identical or similar

results. We used GraphPad Prism 9 (GraphPad Software) for statistical analyses. Quantification was represented by the mean value ± standard deviation for each group. Two-tailed Student $t$ test analyses were performed to compare the mean values between 2 groups. $N$ represents the number of samples in each group. Statistical significances were designated as ns $P > 0.05$, * $P < 0.05$, ** $P < 0.01$, *** $P < 0.001$, and **** $P < 0.0001$.

## Supporting information

**S1 Fig. Identification of GFP11-i insertion sites.** (**A**) AlphaMissense-based impact predictions of missense mutations in human α-tubulin TUBA1A, β-tubulin TUBB2B, KRAS, or BRAF proteins. The impact of mutations are categorized into 3 distinct groups: likely pathogenic (magenta), ambiguous (orange), and likely benign (green). N indicated the number of predicted mutation cases by AlphaMissense. (**B**) Conservatism of all amino acids along TBA-5 and TBB-2 sequences. Conservatism ranged from low (blue) to high (red). Positions of H1 and S2 were indicated above each sequence. GFP11-i insertion sites were indicated by black arrows. (**C**) Sequence alignment of all 9 *C. elegans* α-tubulins at H1-S2 loops. The numbers above sequences indicated residue positions in TBA-5. GFP11 flanked by GS-linker was inserted between Gly43 and Val44 in TBA-5. (**D**) Sequence alignment of all 6 *C. elegans* β-tubulins at H1-S2 loops. The numbers above sequences indicated residue positions in TBB-2. GFP11 flanked by GS-linker was inserted between Lys37 and Gly38 in TBB-2. (**E**) Representative PAE plots for each prediction in Fig 1A. Plots were generated by AlphaFold2 and expected position error (Å) ranged from 0 (blue) to 30 (red). Positions of GFP11 in TBA-5 (GFP11-i) were indicated by orange arrowheads. GFP, green fluorescent protein; PAE, predicted alignment error.
(TIF)

**S2 Fig. Structural models of distinctly tagged TBA-5 after Rosetta relax runs.** (**A**) Structural model of untagged (wild-type) α-tubulin TBA-5 (green) with the TRiC / CCT chaperone complex after Rosetta relax runs. Wild-type *Ce*TBA-5 was employed to substitute the template tubulin in 7TUB model (RCSB PDB database) for relax runs. Initially, substituted tubulin was deposited inside the TRiC / CCT complex, and TRiC / CCT complex was set to be rigid. (**B**) Structural model of internal GFP-tagged α-tubulin TBA-5 (orange) with the TRiC / CCT chaperone complex after Rosetta relax runs. Full-length GFP was inserted into the H1-S2 loop of TBA-5. Then, GFP-tagged TBA-5 was employed to substitute the template tubulin in 7TUB for relax runs. (**C**) Structural model of internal mScarlet-tagged α-tubulin TBA-5 (magenta) with the TRiC / CCT chaperone complex after Rosetta relax runs. Full-length mScarlet was inserted into the H1-S2 loop of TBA-5. Then, mScarlet-tagged TBA-5 was employed to substitute the template tubulin in 7TUB for relax runs. (**D**) Structural model of GFP11-i -tagged α-tubulin TBA-5 (cyan) with the TRiC / CCT chaperone complex after Rosetta relax runs. GFP11-i (GFP11 flanked by GS-linker 3) (colored in pink) was inserted into the H1-S2 loop of TBA-5. Then, TBA-5 (GFP11-i) was employed to substitute the template tubulin in 7TUB for relax runs.
(TIF)

**S3 Fig. Fluorescent tagging strategies for mutated α-tubulin TBA-5 (A19V).** (**A**) Structural illumination of wild-type TBA-5 and TBA-5 (A19V). Ala19 was indicated in yellow, Val19 was indicated in magenta, and Ala232 was indicated in cyan. VDW$_{Overlap}$ represented the overlapped distance of van der Waals radius between protruding carbon in A232 and proximal carbon in A19 or V19, which was calculated using ChimeraX software (see also Materials and methods). Clashed atoms were connected with yellow lines. "+" indicated plus end when TBA-

5 was assembled into MTs. (**B**) Linear representation of distinctly labeled TBA-5 (A19V) constructs for OE. GFP-N, *gfp::3xgs-linker::tba-5*; GFP-C, *tba-5::3xgs-linker::gfp*; GFP11-i, *tba-5(1–43)::gs-linker::gfp11::gs-linker::tba-5(44–447)*. The length of one unit of GS linker in magenta was 6 aa. (**C**) Linear representation of fluorescent labeled TBA-5 (A19V) constructs for KI. Scarlet-i, *tba-5(1–43)::gs-linker 3::mscarlet::gs-linker 3::tba-5(44–447)*; GFP-i, *tba-5(1–43)::gs-linker 3::gfp::gs-linker 3::tba-5(44–447)*. KI, knock-in; MT, microtubule; OE, overexpression. (TIF)

**S4 Fig. Transgenic expression of labeled *tba-5 (A19V)* constructs.** (**A** and **B**) Morphologies and models of amphid (**A**) and phasmid (**B**) cilia in *wild-type* or *tba-5 (A19V)* animals at 15˚C. Cilia were visualized using endogenous OSM-6::GFP. Ciliary bases were indicated by orange arrowheads. Scale bar, 2 μm. (**C** and **D**) Morphologies of amphid (**C**) and phasmid (**D**) cilia in *tba-5 (A19V)* animals with transgenic expression of *tba-5 (A19V)* variants. "Untagged" indicated transgenic expression of untagged TBA-5 (A19V). Cilia were visualized using OSM-6::GFP (for untagged group) or labeled TBA-5 (A19V). The detailed GFP-N, GFP-C or GFP11-i constructs could be found in S3B Fig. All independent OE lines (≥3 lines) in each group exhibited similar phenotype. Ciliary bases were indicated by orange arrowheads. Scale bar, 2 μm. (**E**) Phasmid cilium length in different *tba-5 (A19V)* transgenic lines. Cilia in untagged *tba-5 (A19V)* animals were visualized using OSM-6::GFP. Cilia in other transgenic lines were visualized using labeled TBA-5 (A19V). *N* = 20 for each group. (**F**) Phenotypes of amphid cilia in *tba-5 (A19V)* animals with transgenic expression of TBA-5 (A19V) (GFP11-i). *gfp11-i* constructs without linker, or flanked by GS-linker 1, or GS-linker 2, or GS-linker 3, were overexpressed with *gfp1-10* constructs in *tba-5 (A19V)* animals. The detailed GFP11-i constructs could be found in S3B Fig. Cilia were visualized using endogenous DYF-11::wrmScarlet. All independent lines (≥3 lines) in each group exhibited similar phenotype. Scale bar, 2 μm. (**G**) Percentage of dye-filling positive amphids when endogenous TBA-5 (A19V) were labeled using different tags. *N* = 100 for each group. Numerical data for panels E and G are available in S1 Data. d.s., distal segment; m.s., middle segment; N.A, not applicable; OE, overexpression. (TIF)

**S5 Fig. Morphologies of cilia in *tba-5 (A19V) (scarlet-i)* animals.** (**A**) Phasmid cilium length in *tba-5 (A19V) (gfp11-i)* animals at 15 or 25˚C. *N* = 20 for each group. (**B** and **C**) IFT velocities in cilia of wild-type (**B**) or *tba-5 (gfp11-i)* KI (**C**) animals. Representative kymographs of IFT were shown on the left (DYF-11::wrmScarlet as marker); histograms of IFT velocities were shown on the right. Data were show as mean ± SD; *N* = 80 for each group. Scale bar, 2 μm and 5 s. (**D** and **E**) Morphologies of amphid cilia (**B**) and phasmid cilia (**C**) when endogenous TBA-5 (A19V) were labeled with internal mScarlet (Scarlet-i) or GFP (GFP-i). OSM-6::GFP marked the defective cilia when TBA-5 (A19V) was untagged. Scale bar, 2 μm. (**F**) Phasmid cilium length in *tba-5 (A19V) (scarlet-i)* KI animals at 15 or 25˚C. *N* = 20 for each group. Numerical data for panels A, B, C, and F are available in S1 Data. d.s., distal segment; IFT, intraflagellar transport; KI, knock-in; m.s., middle segment. (TIF)

**S6 Fig. Tissue-specific labeling of endogenous TBB-2.** (**A**) Representative images of endogenous TBB-2 (GFP11-i) with ciliated neuron-specific GFP1-10, or endogenous GFP::TBB-2, in *C. elegans* heads and tails. Fluorescence of TBB-2 (GFP11-i) images were enhanced 6 times when compared to GFP::TBB-2 images. Supposed positions of amphid and phasmid cilia were indicated by orange arrowheads. DYF-11::wrmScarlet marked the sensory cilia. Scale bar, 10 μm. (**B**) DIC (bright field) images corresponding to the head, pharynx, vulva, and tail in Fig 4C. Scale bar, 10 μm. (**C**) Representative images of endogenous TBB-2 (GFP11-i) with

germline-specific GFP1-10, along the whole body of young-adult worm. Scale bar, 100 µm. DIC, differential interference contrast; GFP, green fluorescent protein.
(TIF)

**S7 Fig. GFP11-i as a dynamic MT marker in mitosis and meiosis.** (**A**) (Left) Mitosis of a GSC. Scale bar, 5 µm. (Middle) Stills from mitosis of GSC indicated within orange dashed box (see also S6 Video). (Right) Kymograph of mitosis of GSC indicated with orange dashed box. Scale bar (horizontal), 2 µm; scale bar (vertical), 2 s. (**B**) (Left) Meiosis of a primary spermatocyte (Spc). Scale bar, 5 µm. (Right) Stills from meiosis of primary spermatocyte indicated with orange dashed box (see also S7 Video). This included meiosis I and meiosis II. (**C**) Stills from meiosis of a fertilized oocyte in utero (see also S8 Video). This included meiosis I and meiosis II. Spindles in anaphase I and anaphase II were indicated by white arrowheads. Edges of cells were depicted by orange dashed curves. PB1, the first polar body; PB2, the second polar body; Sp, sperm. Scale bar, 5 µm. (**D**) Stills from embryonic cleavage of a single-cell zygote (P0) (see also S9 Video). "0 min" represented the time just before nuclear envelope breakdown. Edges of cells were depicted by orange dashed curves. AB and P1 were daughter cells in 2-cell embryos. ABa, ABp, EMS, and P2 were daughter cells in 4-cell embryos. Scale bar, 10 µm. GSC, germline stem cell; MT, microtubule; PB, polar body.
(TIF)

**S8 Fig. Astral MT growth rates in single-cell embryos.** (**A**) Representative image of single-cell embryo in anaphase using EBP-2::mNeongreen as markers. A total of 10 frames (400 ms exp.) were projected as a single image to show the paths of EBP-2 dots, as indicated by orange arrowheads (which represented the subset of growing astral MTs). (**B**) Histogram showing the distribution of astral MT growth rates in metaphase and anaphase using EBP-2::mNeongreen as markers. Green lines (metaphase) and magenta lines (anaphase) showed Gaussian fit curves. Quantitative data were shown as mean ± SD. $N = 70$ MTs for each group. (**C**) Representative image of single-cell embryo in anaphase using TBB-2 (GFP11-i) as markers. A total of 10 frames were projected as a single image. Some astral MTs were indicated by blue arrowheads. Scale bar, 5 µm. (**D**) Histogram showing the distribution of astral MT growth rates in metaphase and anaphase using TBB-2 (GFP11-i) as markers. $N = 70$ MTs for each group. (**E**) Representative images for transfection-positive cells when transfected with *htubb8 (gfp11-i)* (100 ng) and *gfp1-10* (200 ng). Scale bar, 10 µm. (**F**) Representative image for immunostaining of acetylated α-tubulin K40 residue (α-K40Ac, magenta) in transfection-positive cells (indicated by orange arrowheads) and transfection-negative cells (gray arrowhead). Scale bar, 10 µm. (**G**) Whole-cell average anti-αK40Ac fluorescence in transfection-positive cells (+) and transfection-negative cells (−). $N = 20$. (**H**) Representative images for mouse oocytes in metaphase I showing localization of mTUBB5-GFP and histone. Meiotic spindles were indicated by orange arrowheads. (**I**) Representative images for mouse oocytes in metaphase I showing localization of mTUBA4A and histone. Meiotic spindles were indicated by orange arrowheads. Scale bar, 10 µm. Numerical data for panels B, D, and G are available in S1 Data. GFP, green fluorescent protein; MT, microtubule.
(TIF)

**S1 Video. GFP::TBB-2 fluorescence in *tbb-1* RNAi-treated *gfp::tbb-2* single-cell zygote.** This video corresponds to Fig 2E (top). Total time: 25 min. Acquired using a spinning-disk confocal microscopy with 30-s intervals between frames. 150 × sped up.
(AVI)

**S2 Video. TBB-2 (GFP11-i) fluorescence in *tbb-1* RNAi-treated *tbb-2 (gfp11-i); glh-1::t2a:: gfp1-10* single-cell zygote.** This video corresponds to Fig 2E (bottom). Total time: 25 min.

Acquired using a spinning-disk confocal microscopy with 30-s intervals between frames. 150 × sped up.
(AVI)

**S3 Video. TBB-2 (GFP11-i) fluorescence of polymerizing MTs in *tbb-2 (gfp11-i)*; P*col-19::gfp1-10* hypodermis.** This video corresponds to Fig 4D (top). Total time: 10 s. Acquired using a spinning-disk confocal microscopy with 200 ms intervals between frames. 2 × sped up.
(AVI)

**S4 Video. TBB-2 (GFP11-i) fluorescence of depolymerizing MTs in *tbb-2 (gfp11-i)*; P*col-19::gfp1-10* hypodermis.** This video corresponds to Fig 4D (middle). Total time: 10 s. Acquired using a spinning-disk confocal microscopy with 200 ms intervals between frames. 2 × sped up.
(AVI)

**S5 Video. TBB-2 (GFP11-i) fluorescence of translocating MTs in *tbb-2 (gfp11-i)*; P*col-19::gfp1-10* hypodermis.** This video corresponds to Fig 4D (bottom). Total time: 10 s. Acquired using a spinning-disk confocal microscopy with 200 ms intervals between frames. 2 × sped up.
(AVI)

**S6 Video. TBB-2 (GFP11-i) (green) and histone-mCherry (magenta) in *tbb-2 (gfp11-i)*; *glh-1::t2a::gfp1-10* mitotic germline.** This video corresponds to S7A Fig. Total time: 3 min 30 s. Acquired using a spinning-disk confocal microscopy with 3-s intervals between frames. 30 × sped up.
(AVI)

**S7 Video. TBB-2 (GFP11-i) (green) and histone-mCherry (magenta) in *tbb-2 (gfp11-i)*; *glh-1::t2a::gfp1-10* spermatheca.** This video corresponds to S7B Fig. Total time: 31 min. Acquired using a spinning-disk confocal microscopy with 10-s intervals between frames. 200 × sped up.
(AVI)

**S8 Video. TBB-2 (GFP11-i) (green) and histone-mCherry (magenta) in *tbb-2 (gfp11-i)*; *glh-1::t2a::gfp1-10* fertilized oocyte.** This video corresponds to S7C Fig. Total time: 20 min. Acquired using a spinning-disk confocal microscopy with 30-s intervals between frames. 150 × sped up.
(AVI)

**S9 Video. TBB-2 (GFP11-i) (green) and histone-mCherry (magenta) in *tbb-2 (gfp11-i)*; *glh-1::t2a::gfp1-10* single-cell zygote.** This video corresponds to S7D Fig. Total time: 24 min. Acquired using a spinning-disk confocal microscopy with 60-s intervals between frames. 240 × sped up.
(AVI)

**S1 Table. *C. elegans* strains in this study.**
(DOCX)

**S2 Table. CRISPR-Cas9 targets in this study.**
(DOCX)

**S3 Table. Plasmids and primers in this study.**
(DOCX)

**S1 Data. Numerical data for all graphs presented in the study.**
(XLSX)

## Acknowledgments

We thank the facilities of Cell Imaging in the Tsinghua University Technology Center for Protein Research for their help. We appreciate Dr. Suhong Xu (Zhejiang Univ.) and Dr. Wei Zou (Zhejiang Univ.) for providing SHX2025 and DUP228 strains. We appreciate Dr. Carsten Janke (Institut Curie) for proofreading this paper. We thank Dr. Dou Wu and Dr. Wenxin Shao for providing plasmids and strains.

## Author Contributions

**Conceptualization:** Kaiming Xu, Guangshuo Ou.

**Data curation:** Kaiming Xu.

**Formal analysis:** Kaiming Xu.

**Funding acquisition:** Kaiming Xu, Guangshuo Ou.

**Investigation:** Kaiming Xu, Zhiyuan Li, Linfan Mao, Zhengyang Guo.

**Methodology:** Kaiming Xu, Zhiyuan Li, Linfan Mao, Zhengyang Guo, Zhe Chen, Yongping Chai, Chao Xie.

**Project administration:** Guangshuo Ou.

**Supervision:** Xuerui Yang, Jie Na, Guangshuo Ou.

**Validation:** Guangshuo Ou.

**Writing – original draft:** Kaiming Xu, Wei Li, Guangshuo Ou.

**Writing – review & editing:** Kaiming Xu.

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
