## [Editor Report · Decision Letter 0]

4 Apr 2024

Dear Dr XU, 

Thank you for submitting your manuscript entitled "Artificial Intelligence-Enabled AlphaFold II Pipeline Guides Functional Fluorescence Labeling of Tubulin Across Species" for consideration as a Methods and Resources Article by PLOS Biology. Please accept my sincere apologies for the long delay in getting back to you as we consulted with an academic editor about your submission. 

Your manuscript has now been evaluated by the PLOS Biology editorial staff, as well as by an academic editor with relevant expertise, and I am writing to let you know that we would like to send your submission out for external peer review.

Once your full submission is complete, your paper will undergo a series of checks in preparation for peer review. After your manuscript has passed the checks it will be sent out for review. To provide the metadata for your submission, please Login to Editorial Manager (https://www.editorialmanager.com/pbiology) within two working days, i.e. by Apr 06 2024 11:59PM.

Kind regards,

Richard

Richard Hodge, PhD

rhodge@plos.org

PLOS

---

## [Decision Letter · Decision Letter 1]

17 Jun 2024

Dear Dr XU,

Thank you for your patience while your manuscript "Artificial Intelligence-Enabled AlphaFold II Pipeline Guides Functional Fluorescence Labeling of Tubulin Across Species" was peer-reviewed at PLOS Biology. It has now been evaluated by the PLOS Biology editors, an Academic Editor with relevant expertise, and by several independent reviewers. I apologize for the delay in coming to this decision.

In light of the reviews, which you will find at the end of this email, we would like to invite you to revise the work to thoroughly address the reviewers' reports.

As you will see below, the reviewers are quite positive, although they do have some concerns and suggestions. The reviewers would like to see more explanation on your developed method, and would like you to include the reasoning behind selecting the specific residue used here. We think that most of the reviewers’ comments are reasonable, although we advise you to not split the manuscript into two (as suggested by R1). Furthermore, while the MAP experiment suggested by R2 and the stable cell line suggested by R3 would undoubtedly strengthen the paper, the Academic Editor considers these to be optional.

Given the extent of revision needed, we cannot make a decision about publication until we have seen the revised manuscript and your response to the reviewers' comments. Your revised manuscript is likely to be sent for further evaluation by all or a subset of the reviewers.

**IMPORTANT - SUBMITTING YOUR REVISION**

*Re-submission Checklist*

*Published Peer Review*

*PLOS Data Policy*

*Blot and Gel Data Policy*

Sincerely,

Suzanne

Suzanne De Bruijn, PhD, 

Associate Editor

PLOS Biology

sbruijn@plos.org

REVIEWS:

*Reviewer #1: This is a very detailed paper with many different experiments, supplemented with computational prediction approaches with AlphaFold II, and fluorescence microscopy.

In essence, this work introduces a amino acid linker system of various length that should facilitate tubulin isotype labeling with GFP, at the inner microtubule lumen. The linker is predicted to avoid clashes with functional MAP interactions in a living cell, an apparent problem with conventical tubulin - GFP tagging. They use human cell lines, murine embryonic tissue, as well as C. elegans as a model system, as it offers many areas of diverse tubulin function in one organism.

Fig. 1: How unique was the structural prediction shown here? This location seems like a very flexible arrangement. Hence, I would expect a large number of possible solutions.

Fig. 2: What is the ratio between of tagged tubulin and endogenous tubulin in each case tested? Was the tagged tubulin the only available variant, or was this a mix with untagged tubulin?

The link to Tubulinopathies is a far distance claim. Obviously, every useful method might add some knowledge to diseases and pathological conditions, but this paper is a structural study into a method improvement (GFP tagging of tubulin) that may, or may not contribute anything.

I am not so convinced that the linker between GFP and tubulin is really necessary to maintain function. GFP tagging has been widely used in the past with no obvious interferences of GFP and tubulin function, mainly because only a fraction of tagged protein is necessary to achieve a good fluorescent signal. Hence, leaving large areas of microtubules free for MAP interactions, inside the barrel, or outside. However, I do not want to diminish the value of this work, but I am confused by the presentation and the lack of focus.

My recommendation to the authors for this paper is, focusing on the methods, and their validation with the necessary experiments (further emphasizing examples where the conventional tagging indeed caused a problem like in embryos), and argue more convincingly about the usefulness of structure prediction software (AlphaFold II in this case for the design of the linker). It seems to me that this paper could easily being split into two papers, a method focused one and one with a focus on the actual biology that is shown here. Currently, this paper is too long, too complex, and therefore hard to read.

I will not say that the data is bad, but the presentation is confusing with over-stuffed figures, and with no strong focus on the apparent outreaching advantages of the methods shown here. This paper could be very interesting to a larger community, but it should be revised to a form that addresses a broad scientific community.

*Reviewer #2: The manuscript entitled "Artificial Intelligence-Enabled AlphaFold II Pipeline Guides Functional Fluorescence Labeling of Tubulin Across Species" elegantly addresses a major challenge in the tubulin field: correctly tagging alpha and beta tubulin isoforms. To date, most tags used to visualise tubulin are placed at the C-terminal or N-terminal region of the protein, which negatively affects its dynamics and proper protein-protein recognition and interaction. To tackle this issue, Kaiming Xu et al. argue that regions within the protein sequence with low conservation are the best option for tagging. They selected the H1-S2 loop in tubulin and began searching for the best option for fluorescently labeling this protein. They performed Rosetta relax runs to determine how the fluorescent protein would affect the folding of tubulin. They discovered that full-length versions of fluorescent proteins negatively impact tubulin folding. Consequently, they decided to use a split version of GFP (GFP11).

Next, they searched for the optimal linker length to attach GFP11 to the H1-S2 loop using AlphaFold II, discovering that, of the three linkers they tested, the best option was the 16 aa (GS) linker. Following this optimization, they demonstrated their improved tubulin tagging method in different models, including both alpha and beta isoforms. They conducted impressive work in C. elegans, showing that this tagging does not affect tubulin properties, unlike current N-terminal and C-terminal tags. They tested overexpression as well as endogenous tagging of different tubulin isotypes and mutated versions, showing that tubulin function remains unaffected in various scenarios. Finally, they applied this method to mammalian tubules from different sources, from human tumor cells to mouse oocytes, demonstrating that this tagging approach is useful for studying the tubulin code in both physiological and pathological environments. Notably, in addition to not affecting tubulin properties, this tagging method provides a higher signal-to-noise ratio, which is very useful in cell work.

In my opinion, the current manuscript represents a significant advancement in protein tagging, particularly for tubulin fluorescent labelling, and it can be a substantial improvement in the study of the tubulin code, focusing on tubulin isotype composition and function in cells. I believe the manuscript should be accepted with minor revisions. Please find some comments and suggestions below:

(i) I would appreciate a basic schematic of the pipeline used to design and improve the tagging in Figure 1. This would greatly aid in understanding the bioinformatics aspect of this work.

(ii) The authors directly label the H1-S2 loop, although they mention that low-conservation regions within the sequence could be good options for tagging. Have they explored other regions? It would be a good control of their pipeline to compare other options with the chosen one, at least in silico.

(iii) Additionally, I miss some in vitro experiments or at least some in silico designs of other tags besides fluorescent proteins. Have the authors considered other tags, such as those used for purification?

(iv) As mentioned in the discussion section, it would be helpful to demonstrate that MAPs binding is not affected by this tagging (they focus on luminal MAPs, but I would check others as well). Have the authors considered purifying and injecting some fluorescent-MAPs to study this?

(v) The units used to measure brood size are missing from the text and Figures 2C and 2D.

*Reviewer #3: The manuscript by Kaiming Xu and colleagues from the laboratory of Guangshuo Ou reports a novel fluorescent labelling strategy of tubulin that improves its polymerization capacities and limits its interference with the co-assembly with other non-labelled tubulins. The strategy is based on numerical simulations of the conformation of tubulins, which take advantage of artificial intelligence to identify the residues to which the labelling would not interfere with protofilament elongation and microtubule assembly. The authors demonstrate that the direct binding of GFP to these residues is incompatible with microtubule elongation. However, they found that the binding of a portion of GFP to the tubulin in parallel with the co-expression of the complementary portion in the cytoplasm is fully functional. The co-expression of the two constructs in the C. elegans embryo led to the formation of dynamic microtubules, which exhibited an improved signal-to-noise ratio when imaged compared to the results obtained using classical strategies. 

Subsequently, the authors demonstrated that the expression of specific isotypes of tubulin in a defined lineage of the worm enables the assembly of fully functional microtubules in the cell body and their cilia. Moreover, the same strategy could be applied to mammalian cells in culture and to mouse embryos. In conclusion, these data are highly promising and compelling, and the presented strategy is likely to be of broad interest, initially in the field of tubulins and microtubules, but also in other areas such as non-perturbing labelling of proteins in living cells and organisms.

There is already a large amount of data in this manuscript which can be published as it is and I don't want to ask more to the authors. I have no expertise in the use of artificial intelligence and don't know much about ciliogenesis in C elegans. However I have some experience in the study of microtubules in mamalian cells in culture. And I think the author are in position to overcome a great limitation in the field which is the generation of stable cell lines expressing a non-perturbing form of fluorescently labelled tubulin. Probably for the reasons of poisoning discussed in the introduction of the manuscript, there is currently no such cell lines. Colleagues have to work on transient transfections or microinjection of tubulin. To my knowledge, no lab has generated a stable cell line in which microtubules would have a proper dynamic. Such a stable cell line would be of great use to the community and would significantly improve the impact of the work. Since the work has been done in HeLa I guess authors could generate a stable line from HeLa but a cell line with a proper control of mitosis and with good ciliogenesis potential would be much more useful. Hence I recommend the author to generate a cell line expressing their two constructs from RPE1 (a non-transformed human cell line often used to study mitosis and ciliogenesis).

---

## [Editor Report · Decision Letter 2]

18 Jul 2024

Dear Dr XU,

Thank you for your patience while we considered your revised manuscript "Artificial Intelligence-Enabled AlphaFold II Pipeline Guides Functional Fluorescence Labeling of Tubulin Across Species" for publication as a Methods and Resources Article at PLOS Biology. This revised version of your manuscript has been evaluated by the PLOS Biology editors and the Academic Editor.

Based on our Academic Editor's assessment of your revision, I am pleased to say that we are likely to accept this manuscript for publication, provided you satisfactorily address the remaining points raised by the Academic Editor (pasted below my signature and labeled 'Comments from the Academic Editor'). During this round of revision, we strongly recommend enlisting the services of a professional editing service or a native-speaking colleague to improve the quality of the English language and writing in the manuscript. In addition, please make sure to address the following data and other policy-related requests that I have provided below (A-F):

(A) We would like to recommend moving the schematic diagram provided in Figure S1F to main Figure 1, as we think this could help the reader.

(B) We would like to suggest the following modification to the title:

"AlphaFold2-guided engineering of split-GFP technology enables labeling of endogenous tubulins across species while preserving function"

(C) You may be aware of the PLOS Data Policy, which requires that all data be made available without restriction: http://journals.plos.org/plosbiology/s/data-availability. For more information, please also see this editorial: http://dx.doi.org/10.1371/journal.pbio.1001797

-Supplementary files (e.g., excel). Please ensure that all data files are uploaded as 'Supporting Information' and are invariably referred to (in the manuscript, figure legends, and the Description field when uploading your files) using the following format verbatim: S1 Data, S2 Data, etc. Multiple panels of a single or even several figures can be included as multiple sheets in one excel file that is saved using exactly the following convention: S1_Data.xlsx (using an underscore).

-Deposition in a publicly available repository. Please also provide the accession code or a reviewer link so that we may view your data before publication. 

Figure 2B-D, 3D, 3J-H, 3L, 5C, 5F, 6C, S4E, S4G, S5A-C, S5F, S8B, S8D, S8G

(D) Please also ensure that each of the relevant figure legends in your manuscript include information on *WHERE THE UNDERLYING DATA CAN BE FOUND*, and ensure your supplemental data file/s has a legend.

(E) Please ensure that your Data Statement in the submission system accurately describes where your data can be found and is in final format, as it will be published as written there. 

(F) Per journal policy, if you have generated any custom code during the course of this investigation, please make it available without restrictions. Please ensure that the code is sufficiently well documented and reusable, and that your Data Statement in the Editorial Manager submission system accurately describes where your code can be found. 

We expect to receive your revised manuscript within three weeks. 

*Published Peer Review History*

*Press*

Kind regards,

Richard

Richard Hodge, PhD

rhodge@plos.org

COMMENTS FROM THE ACADEMIC EDITOR

1. There are some minor grammatical mistakes, e.g.:

(i) Abstract: “Harnessing the power of AlphaFold II pipeline…”. Insert “the” before AlphaFold”

(ii) (line 163) “after relax following Rosetta relax protocol…”. Relaxation?

(iii) (line 450) Our study presented advances in the functional fluorescence. Should be “presents” or “has presented”

(iv) (line 130) “implicating the superiority of our approach” -> “suggesting that our approach is superior to…”

(v) (line 373) “ background noises” -> “background noise”

2. I would recommend using plain language wherever possible to help readers, e.g.:

(i) “cerebral ontogeny” = brain development?

(ii) “mutations in tubulin-coding genes are intrinsically associated with human diseases collectively designated as tubulinopathies”. Does “are intrinsically associated with” mean “cause”?

There are other examples I have not listed

3. AlphaFold II should be AlphaFold 2?

4. I also found that the authors go too far with their language when talking about the utility of their approaches (in places, this reads more like a sales pitch). This is something that remains to be fully seen and I would recommend they use more conservative and objective language on several occasions and let the tools speak for themselves over time.

E.g.

(i) “heralds a transformative paradigm”

(ii) “renders it an indispensable instrument”

There are other examples I have not listed here

---

## [Editor Report · Decision Letter 3]

29 Jul 2024

Dear Dr XU,

On behalf of my colleagues and the Academic Editor, Simon Bullock, I am pleased to say that we can accept your manuscript for publication, provided you address any remaining formatting and reporting issues. These will be detailed in an email you should receive within 2-3 business days from our colleagues in the journal operations team; no action is required from you until then. Please note that we will not be able to formally accept your manuscript and schedule it for publication until you have completed any requested changes.

PRESS

Best wishes, 

Richard

Richard Hodge, PhD

rhodge@plos.org

PLOS
